# Uncertainty quantification in cerebral circulation simulations focusing on the collateral flow: Surrogate model approach with machine learning

**Changyoung Yuhn**[1¤], **Marie Oshima**[2]*, **Yan Chen**[2], **Motoharu Hayakawa**[3], **Shigeki Yamada**[2,4]

**1** Department of Mechanical Engineering, The University of Tokyo, Meguro-ku, Tokyo, Japan, **2** Interfaculty Initiative in Information Studies, The University of Tokyo, Meguro-ku, Tokyo, Japan, **3** Department of Neurosurgery, Fujita Health University, Toyoake, Aichi, Japan, **4** Department of Neurosurgery, Shiga University of Medical Science, Otsu, Shiga, Japan

¤ Current address: Toyota Central R&D Labs., Inc., Nagakute, Aichi, Japan
* marie@iis.u-tokyo.ac.jp

**Data Availability Statement:** All relevant data are available on Zenodo (https://doi.org/10.5281/zenodo.6557448), except for medical images that

## Abstract

Collateral circulation in the circle of Willis (CoW), closely associated with disease mechanisms and treatment outcomes, can be effectively investigated using one-dimensional–zero-dimensional hemodynamic simulations. As the entire cardiovascular system is considered in the simulation, it captures the systemic effects of local arterial changes, thus reproducing collateral circulation that reflects biological phenomena. The simulation facilitates rapid assessment of clinically relevant hemodynamic quantities under patient-specific conditions by incorporating clinical data. During patient-specific simulations, the impact of clinical data uncertainty on the simulated quantities should be quantified to obtain reliable results. However, as uncertainty quantification (UQ) is time-consuming and computationally expensive, its implementation in time-sensitive clinical applications is considered impractical. Therefore, we constructed a surrogate model based on machine learning using simulation data. The model accurately predicts the flow rate and pressure in the CoW in a few milliseconds. This reduced computation time enables the UQ execution with 100 000 predictions in a few minutes on a single CPU core and in less than a minute on a GPU. We performed UQ to predict the risk of cerebral hyperperfusion (CH), a life-threatening condition that can occur after carotid artery stenosis surgery if collateral circulation fails to function appropriately. We predicted the statistics of the postoperative flow rate increase in the CoW, which is a measure of CH, considering the uncertainties of arterial diameters, stenosis parameters, and flow rates measured using the patients' clinical data. A sensitivity analysis was performed to clarify the impact of each uncertain parameter on the flow rate increase. Results indicated that CH occurred when two conditions were satisfied simultaneously: severe stenosis and when arteries of small diameter serve as the collateral pathway to the cerebral artery on the stenosis side. These findings elucidate the biological aspects of cerebral circulation in terms of the relationship between collateral flow and CH.

could potentially identify or reveal sensitive patient information.

**Funding:** This study was supported by the Japan Society for the Promotion of Science through Grants-in-Aid for Scientific Research, Grant Nos. 16H04264 (MO) and 18J21374 (CY). The funder had no role in study design, data collection and analysis, decision to publish, or preparation of the manuscript.

**Competing interests:** The authors have declared that no competing interests exist.

## Author summary

Cerebral arteries generate a ring-like network that provides alternative routes for blood supply in the case of carotid artery stenosis. This collateral circulation is closely associated with the potential risk of stroke and treatment outcomes in patients with stenosis. In this study, we propose a method to elucidate the cerebral circulation of individual patients using a blood flow simulation that incorporates the patient's clinical data. A key feature of our approach is its capability to obtain the probability of the different outputs using simulation, considering the uncertainty of patient conditions. Although this capability is essential for obtaining reliable results, the process is time-consuming and requires numerous computer resources. We solved this problem by combining the blood flow simulation with machine learning to perform predictions 43 000 times faster than conventional simulations. We applied the proposed method to predict cerebral circulation following surgery in three patients with stenosis and verified that the method can assess the surgical risk almost in real-time, even on a desktop computer. Additionally, extensive prediction results (100 000 cases for each patient) obtained using this method clarify the relationship between collateral circulation and life-threatening surgical outcomes.

## Introduction

Carotid artery stenosis is a major risk factor for stroke, one of the leading causes of death and disability worldwide. Stroke can occur if the stenosis reduces the blood supply to the brain significantly. In general, the severity of stenosis is a principal indicator closely associated with the risk of stroke [1]. Despite the severity of the stenosis, most patients are asymptomatic owing to adequate collateral circulation. Collateral circulation refers to the flow of blood through an arterial network connecting the diseased and normal sides and is particularly abundant in cerebral arteries that form a ring-like network. If collateral circulation is adequate, cerebral blood flow is maintained regardless of stenosis. However, if the severity of the stenosis increases such that the collateral circulation attains its limit or if certain arteries are absent in the patient, cerebral blood flow on the diseased side is no longer maintained. In such cases, the best opportunity for treatment is lost by the time the patient develops symptoms.

For predictive medicine, such as stroke prediction, a hemodynamic simulation is a promising tool that provides clinically relevant hemodynamic quantities under various conditions. However, the clinical application of simulation tools has certain requirements. First, as stroke is associated with aging or arteriosclerosis [2], the simulation should reflect the effects of these factors on the entire cardiovascular system. Second, the computation time must be sufficiently short to obtain immediate clinical feedback of the simulation results. To satisfy these requirements, a one-dimensional–zero-dimensional (1D–0D) model is considered practical for simulations. The 1D–0D model is multi-scale and considers the entire cardiovascular system. The model can capture the systemic effects of local arterial changes and thus reproduce hemodynamics that reflects biological phenomena in vivo. Additionally, it facilitates rapid assessment of the primary features of blood flow, such as flow distribution and pulse wave propagation in the arterial network [3–6]. On comparing the 1D–0D model with typical three-dimensional (3D) simulations [7,8], in vitro measurements [9,10], and in vivo measurements [3,11,12], it was observed that the 1D–0D model provides accurate results for spatially averaged flow rate and pressure. It has also been widely used to answer specific clinical questions on hemodynamics in cerebral [13], hepatic [14], and visceral [15] arteries.

Furthermore, the 1D–0D simulation is suitable for investigating the collateral circulation in cerebral arteries. Over the past decade, the 1D–0D model of cerebral circulation has been extensively developed. Cerebral arteries have been modeled in detail, including the circle of Willis (CoW), which serves as a major collateral pathway [11,13,16]. Changes in cerebral circulation caused by arterial occlusion [16], cerebral autoregulation [17], and surgeries for carotid artery stenosis [18,19] have been increasingly investigated. Moreover, recent studies have focused on individualizing models by incorporating patients' clinical data. Typically, this patient-specific approach uses geometric data obtained from medical imaging, such as computed tomography (CT) or magnetic resonance imaging (MRI), to assign parameters and assimilate the measurements of flow and pressure into the simulation [20–22]. Such individualized simulations directly reflect the patients' physiological condition in their predictions, thus yielding precise outputs.

However, simulation-based predictions are often restricted by their deterministic nature, wherein output quantities do not account for uncertainties in clinical data because of the limitations in existing measurement techniques. Uncertainties in clinical data are generated from various sources, including limited resolution, threshold-based lumen segmentation, measurement errors, and fluctuations in blood flow. Particularly, obtaining small diameters of cerebral arteries from medical images with limited resolution involves considerable uncertainty. Moreover, as cerebral arteries are surrounded by the skull, flow measurements are often subjected to severe limitations, and the measured values exhibit large variations. Such uncertainties change the geometric and physiological parameters when incorporated into the simulation, thereby affecting the output significantly. Therefore, it is essential to assess the variability of simulation outputs caused by uncertainties to obtain reliable results for effective decision-making.

Typically, stochastic approaches to uncertainty quantification (UQ), such as Monte Carlo methods, require numerous simulations to obtain the output statistics. This inevitably increases the required computational cost for UQ, posing a major challenge to its implementation in a practical clinical setting. Therefore, several studies on hemodynamic simulation have focused on two primary strategies to reduce the cost of UQ to a feasible scale. The first strategy involves reducing the number of required simulations. Herein, the core idea is to explore the stochastic space efficiently using stochastic collocation methods [23] and multi-resolution stochastic expansion [24,25] to achieve faster convergence of statistics. The second strategy involves reducing the cost of an individual simulation by employing a 1D–0D model [26–28]. Despite the considerable progress reported in recent studies, implementing UQ in routine clinical diagnosis remains a challenge. As individual simulations generally involve iterative calculations to assimilate data or obtain converged solutions, even the UQ based on 1D–0D model is intractable in medical institutions, where time and computational resources are often limited.

This problem can be addressed by constructing a data-driven surrogate model, obtained by fitting a regression model to the simulation data. The surrogate model performs predictions based on the superficial input–output relationships of well-established cardiovascular models, which significantly accelerate the predictions while maintaining accuracy. In the context of data-driven modeling, machine learning with deep neural networks (DNNs) has been widely explored in recent years [29,30]. Although integrating machine learning techniques with hemodynamic simulations has been actively researched in the past few years [31–34], most studies focused on predicting the fractional flow reserve in coronary arteries.

In this study, we constructed a surrogate model of the cerebral circulation to replace the existing 1D–0D simulation. This resulted in the fast execution of UQ (within a few minutes) even on a desktop computer. We used the surrogate model to perform the UQ for investigating

the biology of cerebral circulation, focusing on collateral circulation through the CoW. Particularly, we predicted the risk of cerebral hyperperfusion (CH) and analyzed its relationship with collateral circulation. Similar to stroke, CH is considered to occur when collateral circulation fails to function appropriately [35–38]. CH is defined as an increase of more than 100% in the time-averaged flow rate through the cerebral arteries immediately after carotid artery stenosis surgery as compared to preoperative values. Although the incidence varies (0.2–18.9%) [38], CH can lead to intracerebral hemorrhage, which can be life-threatening as indicated by its high mortality rate (38.2%) [39]. Therefore, it is important to identify the patients at risk in the preoperative stage to adopt appropriate interventions for preventing hemorrhages caused by CH [36,40].

For this, we considered three patients with internal carotid artery (ICA) stenosis and predicted the flow rate increase in the cerebral arteries when the stenosis was virtually dilated. The predictions considered the uncertainties in the clinical data used to set the patient-specific conditions. We focused on uncertainties in arterial diameters, stenosis parameters, and flow rates derived from the patients' clinical data. Initially, the uncertainty of these parameters was estimated. Subsequently, the statistics of the flow rate increase under the uncertainty were evaluated through UQ. In addition to the UQ, we performed sensitivity analysis (SA) to measure the impact of each parameter on the flow rate increase. Based on the analysis of the UQ and SA results, we explored the risk factors associated with CH, particularly those related to the collateral circulation function.

## Methods

### Ethics statement

We used the clinical data of actual patients to

- infer a physiologically reasonable range of inputs (in the "Learning data generation" subsection),

- verify the surrogate model (in the "Machine learning" subsection), and

- perform UQ in predicting postoperative CH (in the "Uncertainty quantification and sensitivity analysis" subsection).

Seven patients who underwent endarterectomy or stenting for ICA stenosis were included in the study (Table 1). The imaging data, measurements of inflows and outflows of the CoW, and mean arterial pressure in the upper arm were collected for all patients before surgery. Patient data were collected and provided in an anonymized form by the Rakuwakai Otowa Hospital (Kyoto, Japan) and Fujita Health University Hospital (Aichi, Japan), with written informed consent from the patients. Ethical approval for this study was granted by the Research Ethics Committee of The University of Tokyo, the Ethics Committees for Human Research of Rakuwakai Otowa Hospital, and the Ethics Review Committee of Fujita Health University.

### Overview

In this study, we used the 1D–0D simulation to generate a large dataset comprising a pair of anatomical and physiological conditions (inputs) and the corresponding cerebral circulation under those conditions (outputs). The generated data were used to perform supervised learning of the DNN (Fig 1). This facilitated the construction of a surrogate model that can rapidly predict cerebral circulation under specified anatomical and physiological conditions. The surrogate model was verified by comparing the prediction results of the test data (not used for the training) and actual patient conditions with those obtained from the 1D–0D simulation.

**Table 1. Patient characteristics.**

| Patient | Age/Sex | Lesion[a] | Treatment | Imaging | Flow measurements |
|---|---|---|---|---|---|
| 1 | 82/M | RICA stenosis (59%) | CEA | CT | PC-MRI, SPECT |
| 2 | 63/M | LICA stenosis (83%) | Staged CAS | CT | US, SPECT |
| 3 | 72/M | RICA stenosis (91%) | CEA | CT | PC-MRI, SPECT |
| 4 | 63/M | RICA stenosis (35%) | CEA | CT | PC-MRI, SPECT |
| 5 | 68/M | LICA stenosis (63%), RICA stenosis (65%) | CEA | CT | PC-MRI, SPECT |
| 6 | 70/M | LICA stenosis (73%) | CAS | CT/MRI[b] | PC-MRI, SPECT |
| 7 | 79/F | LICA stenosis (86%) | CEA | CT/MRI[b] | PC-MRI, SPECT |

[a]Stenosis ratio in parentheses denotes the percentage reduction in diameter to the maximum distal diameter.

[b]CT scan of the neck and MRI scan of the head.

CAS, carotid artery stenting; CEA, carotid endarterectomy; CT, computed tomography; LICA, left internal carotid artery; MRI, magnetic resonance imaging; PC-MRI, phase contrast magnetic resonance imaging; RICA, right internal carotid artery; SPECT, single photon emission computed tomography; US, ultrasound.

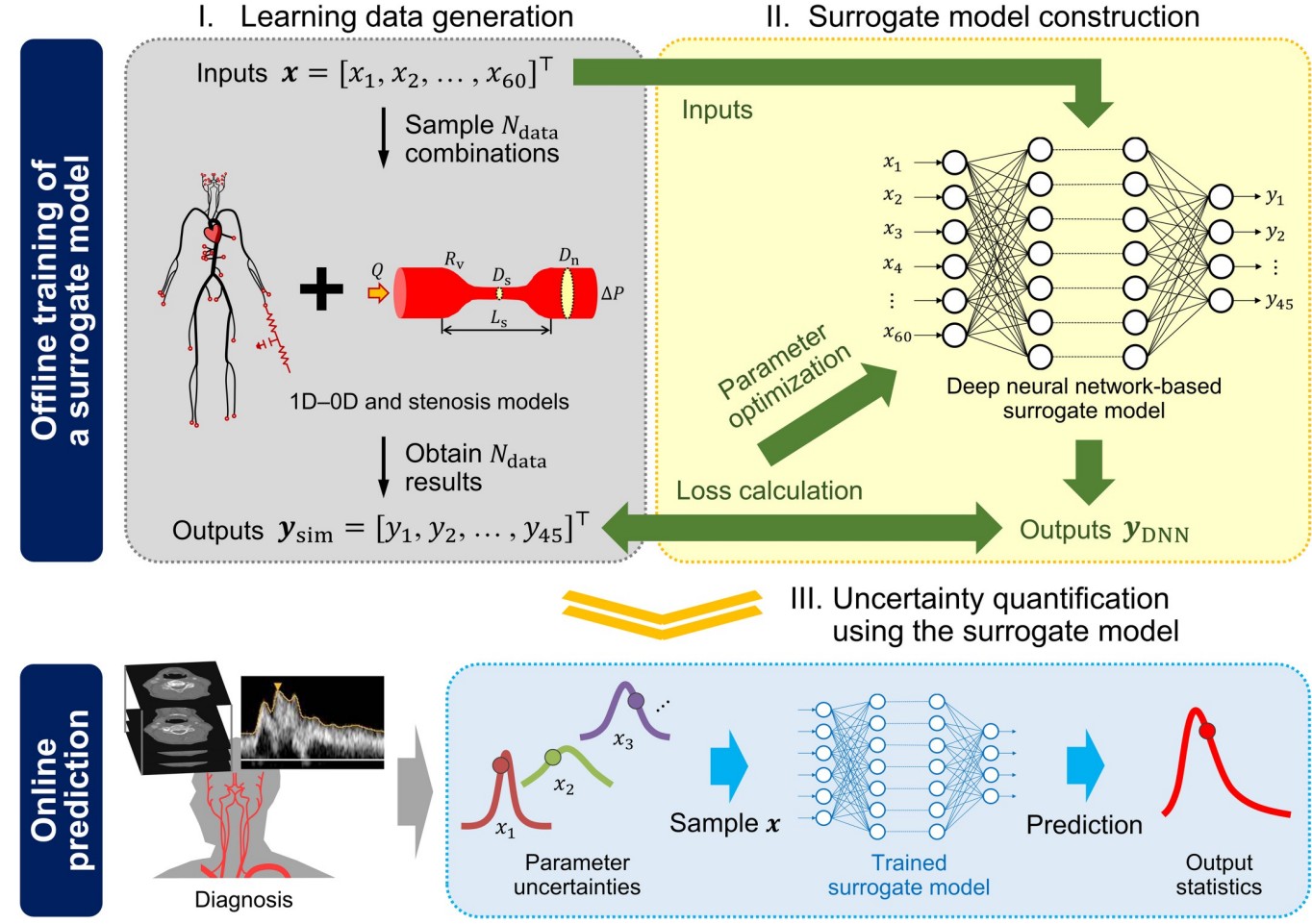

**Fig 1. Overview of the proposed approach to perform uncertainty quantification.** We trained a deep neural network using the datasets obtained from one-dimensional–zero-dimensional (1D–0D) simulation. The datasets were generated by randomly sampling 60 inputs (column vector $\boldsymbol{x} \in \mathbb{R}^{60}$) describing the geometry of cerebral arteries and stenoses, and collecting the corresponding 45 simulation outputs (column vector $\boldsymbol{y}_{\text{sim}} \in \mathbb{R}^{45}$) of time-averaged flow rates and pressures. After performing the data acquisition and model training in the offline phase, the surrogate model was used in the online phase to predict the outputs rapidly. This ensured a fast and efficient uncertainty quantification.

Using the surrogate model as an alternative to the 1D–0D simulation, we performed UQ to predict the percentage increase in the cerebral blood flow caused by the ICA stenosis surgery. We considered the uncertainties in the input parameters derived from the patient's clinical data, including the arterial diameters, stenosis parameters, and flow rates. The possible range of each uncertain parameter was defined, and the uncertainties were subsequently propagated using the Monte Carlo method. Additionally, SA was conducted to quantify the impact of each parameter on the predicted results.

The four primary segments of the methods used in this study include the 1D–0D simulation, learning data generation, machine learning, and UQ and SA. The remaining subsections focus on the details of the method for each segment.

## 1D–0D simulation

We employed the closed-loop 1D–0D cardiovascular model developed by Liang et al. [18,41] for blood flow simulations. In this model, large arteries are represented as 1D segments, which are assumed to be straight, axisymmetric, and deformable tubes. The arterial network comprises 83 segments that contribute to the systemic circulation throughout the body, including 22 segments of the cerebral circulation, as depicted in Fig 2. The inlet and outlet boundary conditions for the 1D network were obtained by coupling the network with the 0D closed-loop model, which represents the peripheral circulation and heart as lumped parameter networks. In the subsequent subsections, we briefly explain the governing equations of the models, numerical methods used to solve them, and the methods implemented for the patient-specific setup of the simulation.

**Governing equations.** The governing equations for blood flow in 1D arteries are derived from the principle of conservation of mass and momentum, as follows [5,6]:

$$\frac{\partial A}{\partial t} + \frac{\partial Q}{\partial x} = 0, \tag{1}$$

$$\frac{\partial Q}{\partial t} + \frac{\partial}{\partial x}\left(\frac{Q^2}{A}\right) + \frac{A}{\rho}\frac{\partial P}{\partial x} = -K_R \frac{Q}{A}. \tag{2}$$

Herein, $t$ represents the time; $x$ indicates the axial coordinate along the artery; $A$, $Q$, and $P$ denote the cross-sectional area of the artery, volumetric flow rate, and internal pressure, respectively; $\rho$ = 1060 kg m$^{-3}$ indicates the blood density; and $K_R = 22\pi\mu/\rho$ represents the resistance parameter [4] with blood viscosity $\mu$ = 0.0047 Pa s. The ($A$, $Q$) system in Eqs (1) and (2) is closed by the relationship between the pressure and cross-sectional area, derived from Laplace's law [3, 5] as follows:

$$P - P_0 = \frac{\sqrt{\pi}Eh}{A_0(1 - \sigma^2)}\left(\sqrt{A} - \sqrt{A_0}\right), \tag{3}$$

where $A_0$ denotes the cross-sectional area at reference pressure $P_0$ = 85 mmHg, $h$ indicates the arterial wall thickness, $E$ represents Young's modulus, and $\sigma$ denotes Poisson's ratio. In this study, $\sigma$ was set to 0.5, and $Eh$ in Eq (3) was assigned based on the empirical relationship with the arterial radius [3].

In a stenotic artery, the abrupt changes in the cross-sectional area cause a large pressure loss associated with flow separation and reattachment. As the 1D model alone cannot completely describe such a pressure loss, a stenosis model, which relates pressure loss across the stenosis ($\Delta P$) to geometric parameters, was coupled with the 1D model. We employed the

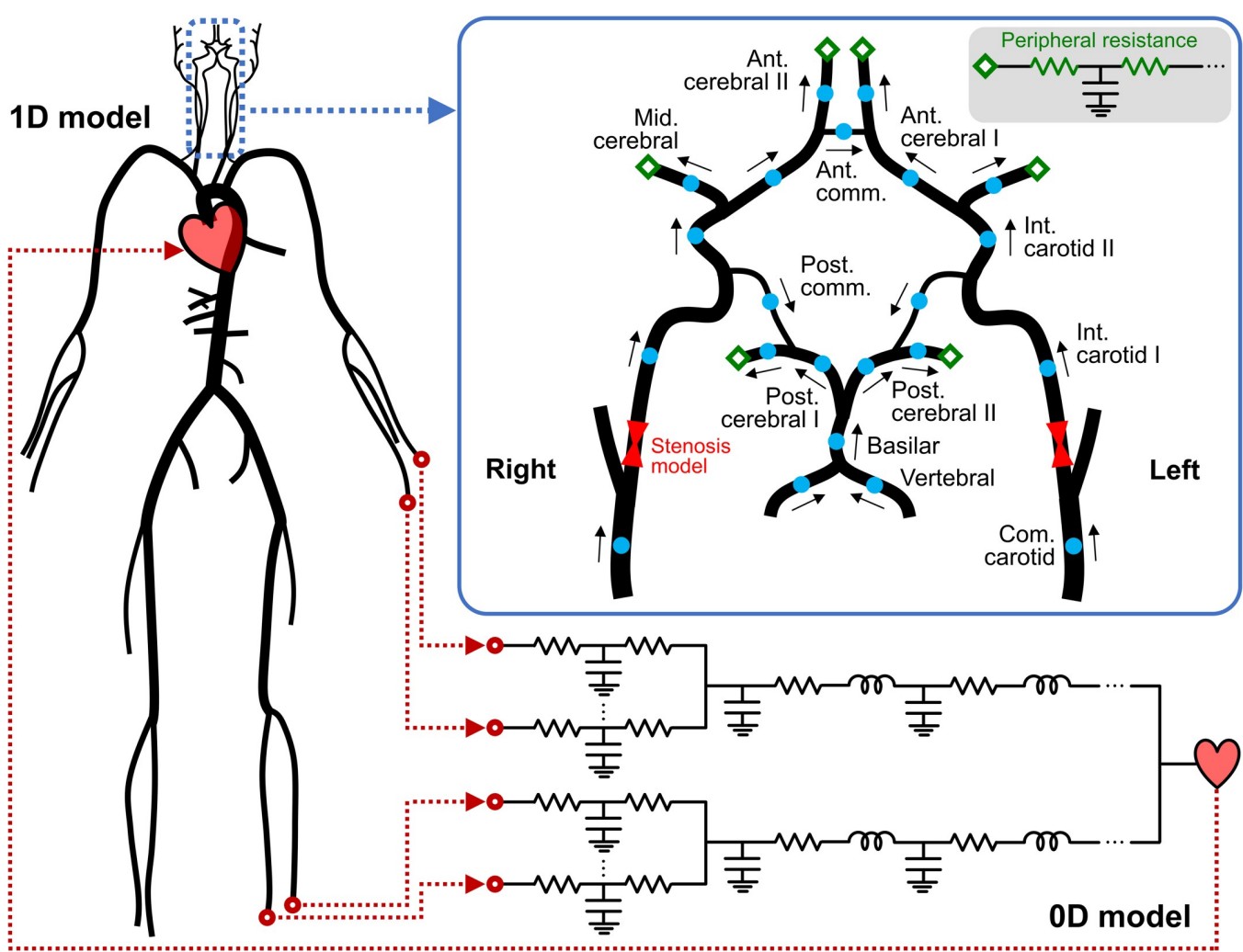

**Fig 2. Schematic representation of the one-dimensional–zero-dimensional (1D–0D) model.** The 1D network comprises 83 arterial segments, including 22 segments (blue dots) composing the cerebral circulation. Cerebral arteries form a ring-like network, referred to as the circle of Willis, which supplies blood to the brain through the six outlets (green diamonds). The arrows indicate the direction of the flow defined as positive in the simulation. The inlet and outlet boundary conditions for the 1D network are obtained by coupling with the 0D closed-loop model, which represents the peripheral circulation and heart.

model reported in previous studies [42–44]:

$$\Delta P = R_{\mathrm{v}} Q + K_{\mathrm{t}} \frac{8\rho}{\pi^2 D_{\mathrm{n}}^4} \left\{ \frac{1}{(1 - SR)^2} - 1 \right\}^2 Q|Q| + K_{\mathrm{u}} \frac{4\rho L_{\mathrm{s}}}{\pi D_{\mathrm{n}}^2} \dot{Q}, \tag{4}$$

$$R_{\mathrm{v}} = \int_0^{L_{\mathrm{s}}} \frac{128\mu}{\pi \{D(x)\}^4} \, dx, \tag{5}$$

where $R_{\mathrm{v}}$ denotes the viscous resistance of the stenosis, evaluated considering the axial diameter change $D(x)$; $D_{\mathrm{n}}$ indicates the maximum diameter distal to the stenosis; $SR$ represents the stenosis ratio defined as the percentage reduction in diameter ($1 - D_{\mathrm{s}}/D_{\mathrm{n}}$, with the minimum stenosis diameter $D_{\mathrm{s}}$); $L_{\mathrm{s}}$ denotes the stenosis length; and $\dot{Q}$ indicates the time derivative of $Q$.

The first, second, and third terms in Eq (4) account for the contribution of viscous friction, flow separation, and pulsatility to the pressure loss, respectively. Although coefficients $K_t$ and $K_u$ rely on the stenosis geometry, they have been empirically set to 1.52 and 1.2, respectively, in the literature [43,45]. In this study, we considered $K_t$ as an uncertain parameter ranging between 1.0 and 2.699 [46], whereas $K_u$ was maintained constant at 1.2, owing to its negligible influence on $\Delta P$.

The 0D closed-loop model comprises the peripheral artery, upper and lower body blocks, and heart (Fig 2). The peripheral arteries distal to the 1D terminal arteries are represented by the three-element Windkessel model (RCR circuit). In each upper or lower body block, the capillaries, venules, and veins are modeled as RLC circuits in series. The heart is modeled based on the time-varying elastance method, which provides the inlet boundary condition to the 1D network, generating a closed-loop system. The governing equations for the 0D model are derived by linearizing and integrating Eqs (1)–(3) along the axial direction, as follows [47,48]:

$$C\frac{dP_1}{dt} + Q_2 - Q_1 = 0, \tag{6}$$

$$L\frac{dQ_2}{dt} + P_2 - P_1 = -RQ_2, \tag{7}$$

where $R$, $L$, and $C$ represent the viscous resistance, inertia of blood, and vascular compliance, respectively; and subscripts 1 and 2 denote the quantities upstream and downstream, respectively.

**Numerical methods.** The governing equations for the 1D model were solved using the two-step Lax–Wendroff scheme. Bifurcated 1D segments were coupled by enforcing the conservation of mass and total pressure at the bifurcations. As this yields the coupled nonlinear equations (see [41] for detailed formulas), we used the iterative Newton–Raphson method to solve them [6,41]. Furthermore, simultaneous ordinary differential equations in the 0D model were solved using the fourth-order Runge–Kutta scheme. The 1D, 0D, and stenosis models were coupled using Riemann invariants [41].

**Patient-specific modeling.** Initially, all model parameters were assigned as reported by Liang et al. [18,41]. Subsequently, certain parameters associated with cerebral circulation were assigned or adjusted based on the patient's clinical data. The patient-specific parameters included the diameters and lengths of the carotid and cerebral arteries, stenosis model parameters, and peripheral resistances (PRs), which represent the sums of the two resistances in the three-element Windkessel model at the six outlets of the CoW. Additionally, the stiffness and diameter of the aorta were adjusted based on the patient's age, and the total PR was adjusted to match the measured pressure.

The diameters and lengths of the carotid and cerebral arteries were extracted from medical images (CT or MRI) and directly assigned to the corresponding parameters in the 1D model. In the case of the stenotic artery, $R_v$, $D_n$, $SR$, and $L_s$ in Eqs (4) and (5) were evaluated based on the acquired geometry. Image processing for arterial lumen segmentation, centerline extraction, 3D reconstruction, and calculation of geometric parameters was conducted using in-house software, namely "V-Modeler" [49]. As routine diagnostic imaging generally involves only the diseased region (head and neck in this case), patient-specific geometries for the remaining 1D segments could not be obtained. Therefore, we used the geometries prescribed in the literature for these segments; however, the stiffness and diameters of the aortic segments were modified to reflect age-related changes [50].

At the six outlets of the CoW, including the left and right anterior, middle, and posterior cerebral arteries, the PRs were adjusted through iterative calculations to match the measured flow rates [20,51]. Outflow rates in these arteries were measured using single photon emission computed tomography (SPECT) combined with phase contrast magnetic resonance imaging (PC-MRI) or ultrasound measurement [20]. Initially, we converted the regional brain perfusion map on SPECT images to the flow rates averaged over a cardiac cycle duration at the six outlets, $\{\bar{Q}_i^{\text{SPECT}}\}_{i=1}^6$, using vascular territory templates [52]. Subsequently, these flow rates were corrected as follows based on the measured total inflow rate to the CoW while maintaining constant flow distribution ratios among the outlets:

$$\bar{Q}_i^{\text{target}} = \bar{Q}_{\text{total}} \cdot \frac{\bar{Q}_i^{\text{SPECT}}}{\sum_{j=1}^6 \bar{Q}_j^{\text{SPECT}}}, \quad i = 1, 2, \ldots, 6. \tag{8}$$

Herein, $\bar{Q}_{\text{total}}$ denotes the summation of the flow rates in the three inlets of the CoW (the left and right ICAs and the basilar artery), which is measured either using PC-MRI or ultrasound. Finally, we used $\bar{Q}_i^{\text{target}}$ in Eq (8) as the target value for the PR adjustment.

The total PR was adjusted to match the patient's mean arterial pressure measured at the upper arm [51]. This was implemented by changing the PRs of the terminal arteries, excluding the CoW, with the scaling factor relative to the initial values.

## Learning data generation

We generated a dataset of simulated cerebral circulation for 200 000 synthetic conditions using the 1D–0D simulation. These conditions reflected the anatomical and physiological variations in patients with and without ICA stenosis and were reproduced by randomly sampling 60 input parameters within a reasonable range (as will be discussed later in the "Design of experiments" subsection). The dataset was used for the supervised learning of the DNN. The following subsections describe the steps for generating the learning data, which include defining inputs and outputs, designing the input space for collecting the data samples, and running simulations.

**Defining inputs and outputs.**   Although the 1D–0D model includes a large number of parameters, only some have a significant impact on cerebral circulation. As described in the "Patient-specific modeling" subsection, we set those parameters in a patient-specific manner based on the patients' clinical data. The parameters include

- Diameters of 22 carotid and cerebral arteries in the 1D model (22 parameters);

- Lengths of 22 carotid and cerebral arteries in the 1D model (22 parameters);

- $R_v$, $D_n$, and $SR$ in Eq (4) for each left and right ICA stenoses (6 parameters);

- PRs at the six outlets of the CoW (6 parameters);

- Scaling factor for the total PR (1 parameter);

- Age (1 parameter);

and an uncertain parameter for the stenosis model, which is

- $K_t$ in Eq (4) for each left and right ICA stenoses (2 parameters).

The aforementioned 60 parameters characterize the patient's anatomical and physiological conditions and significantly affect the cerebral circulation. We selected all these parameters as inputs to capture their influence on the cerebral circulation. Note that we do not select the

stenosis length, $L_s$, as an input. As shown in Eq (4), $L_s$ has two effects on $\Delta P$: one on the third term and the other on the first term via $R_v$. The effect of $L_s$ on the third term can be ignored because the third term is negligible compared to the other terms. Furthermore, since we selected $R_v$ as the input representative of the stenosis geometry, encompassing the variations of $D(x)$ and $L_s$, it is unnecessary to select $L_s$ as a separate input to be varied.

Based on the 1D–0D simulation, $A(t, x)$, $Q(t, x)$, and $P(t, x)$ at each axially aligned grid point of the 1D artery can be obtained as the output. However, according to the definition of CH (percentage increase in time-averaged flow rate), the focus lies on the assessment of cerebral circulation as a "time average" in several clinical situations. Therefore, we aimed to construct a surrogate model that predicts hemodynamic quantities averaged over a cardiac cycle duration and limits the outputs to be predicted, which include

- Cycle-averaged flow rates, $\bar{Q}$, in the middle of the carotid and cerebral arteries (22 quantities);

- Cycle-averaged pressures, $\bar{P}$, in the middle of the carotid and cerebral arteries (22 quantities);

- Mean arterial pressure, which is the cycle-averaged pressure in the middle of the left subclavian artery (1 quantity).

Here, cycle-averaged flow rate and pressure refer to $Q$ and $P$ averaged over a cardiac cycle duration:

$$\bar{Q}(x) = \frac{1}{T_c} \int_{t_s}^{t_s + T_c} Q(t, x) dt, \tag{9}$$

$$\bar{P}(x) = \frac{1}{T_c} \int_{t_s}^{t_s + T_c} P(t, x) dt, \tag{10}$$

where $t_s$ and $T_c$ respectively denote the time to start averaging and cardiac cycle duration (fixed as 1 s). In the axial direction, $\bar{Q}$ is constant and $\bar{P}$ decreases almost linearly unless there is a significant axial change in $\bar{A}$. Therefore, $\bar{Q}$ and $\bar{P}$ at the middle grid point of each artery can be regarded as the axially averaged quantities in each artery. The aforementioned 45 outputs are the primary clinically relevant quantities describing the cerebral circulation. Consequently, we constructed a surrogate model that defines a mapping from the inputs $x \in \mathbb{R}^{60}$ to the outputs $y \in \mathbb{R}^{45}$.

**Design of experiments.** The input–output paired learning data can be obtained by randomly sampling $x \in \mathbb{R}^{60}$ and performing 1D–0D simulations to obtain the corresponding $y \in \mathbb{R}^{45}$. In this step, the sampling ranges for $x$ must be adequately prescribed. If the ranges are extremely narrow, the trained surrogate model would be accurate only in limited input space, restricting the model's coverage. Particularly, the prediction accuracy of the DNN outside the trained range decreases significantly because of its interpolative nature [53]. Therefore, we inferred physiologically reasonable ranges for $\{x_n\}_{n=1}^{60}$ by investigating the data of the seven patients (Table 1) and reviewing the literature [54–58]. The basic policy was to calculate the mean and standard deviation (SD) of the data to adopt a range that covers the mean ± 3SD (details in S1 Appendix). Table 2 summarizes the ranges of inputs used to generate the learning data.

**Simulations.** Learning data were generated considering four scenarios, namely (i) intact ICAs, (ii) left ICA stenosis, (iii) right ICA stenosis, and (iv) left and right ICA stenoses. In each scenario, $x$ was randomly sampled in the prescribed range (Table 2); however, the stenosis

**Table 2. Sampling ranges of inputs used to generate the learning data.**

| Input parameters | | Ranges | | |
|---|---|---|---|---|
| Diameter (mm), length (mm), peripheral resistance (mmHg s mL$^{-1}$) | R. com. carotid | [3.9, 11.6], | [78, 222], | — |
| | L. com. carotid | [3.9, 11.6], | [109, 252], | — |
| | R. int. carotid I | [2.3, 6.8], | [120, 195], | — |
| | L. int. carotid I | [2.3, 6.8], | [120, 195], | — |
| | R. int. carotid II | [1.9, 6.0], | [2, 12], | — |
| | L. int. carotid II | [1.9, 6.0], | [2, 12], | — |
| | R. vertebral | [1.4, 4.9], | [113, 276], | — |
| | L. vertebral | [1.4, 4.9], | [113, 276], | — |
| | Basilar | [1.6, 4.9], | [15, 36], | — |
| | R. ant. cerebral I | [0.1[a], 3.6], | [7, 31], | — |
| | L. ant. cerebral I | [0.1[a], 3.6], | [7, 31], | — |
| | R. ant. cerebral II | [1.2, 3.6], | [6, 45], | (0, 200] |
| | L. ant. cerebral II | [1.2, 3.6], | [6, 45], | (0, 200] |
| | R. mid. cerebral | [1.4, 4.3], | [10, 51], | (0, 100] |
| | L. mid. cerebral | [1.4, 4.3], | [10, 51], | (0, 100] |
| | R. post. cerebral I | [0.1[a], 3.2], | [2, 23], | — |
| | L. post. cerebral I | [0.1[a], 3.2], | [2, 23], | — |
| | R. post. cerebral II | [1.1, 3.2], | [2, 54], | (0, 250] |
| | L. post. cerebral II | [1.1, 3.2], | [2, 54], | (0, 250] |
| | Ant. comm. | [0.1[a], 2.6], | [2, 7], | — |
| | R. post. comm. | [0.1[a], 2.7], | [4, 27], | — |
| | L. post. comm. | [0.1[a], 2.7], | [4, 27], | — |
| Scaling factor for the total peripheral resistance (-) | | [0.5, 2.0] | | |
| Viscous resistance of the stenosis $R_{\mathrm{v}}$ (mmHg s mL$^{-1}$) | | [0, min($R_{\mathrm{v,max}}$, 500)[b]] | | |
| Maximum diameter distal to the stenosis $D_{\mathrm{n}}$ (mm) | | [2.9, 7.0] | | |
| Stenosis ratio $SR$ (%) | | [0, 100) | | |
| Coefficient of the second term in Eq (4) $K_{\mathrm{t}}$ (-) | | [1.0, 2.699] | | |
| Age | | [25, 90] | | |

[a]Missing arteries were represented as extremely narrow arteries with diameters of 0.1 mm, which enabled efficient execution of simulations without requiring a redefinition of the arterial network topology in each case.

[b]The upper bound was defined as a function of $SR$ as $128\mu L_{\mathrm{s,max}}/\pi D_{\mathrm{n,min}}^{4}(1 - SR)^{4}$ with an upper limit of 500 mmHg s mL$^{-1}$ (S1 Appendix). Herein, $L_{\mathrm{s,max}}$ denotes the maximum value of the stenosis length (assumed to be 40 mm), and $D_{\mathrm{n,min}}$ indicates the lower bound of $D_{\mathrm{n}}$.

parameters for the intact ICA were set as $R_{\mathrm{v}} = 0$, $D_{\mathrm{n}} = D_{\mathrm{ICA}}$ (diameter of the ICA), $SR = 0$, and $K_{\mathrm{t}} = 0$. We sampled 50 000 sets of $\boldsymbol{x}$ for each scenario and obtained the corresponding $\boldsymbol{y}$ from the simulation. Consequently, learning data $\{\boldsymbol{x}^{(s)}, \boldsymbol{y}^{(s)}\}_{s=1}^{N_{\mathrm{data}}}$ were generated with the number of samples $N_{\mathrm{data}} = 200\,000$. We observed that certain inputs resulted in unphysical or non-physiological outputs, such as $\bar{P} < 0$, or reversed flow in terminal arteries. Such samples were replaced with new samples. All simulations were performed on the Oakforest-PACS supercomputer system provided by the Information Technology Center at The University of Tokyo (Tokyo, Japan). The samples were equally allocated to 31 280 CPU cores (Intel Xeon Phi 7250). The total computation time required was approximately 25 h.

**Data splitting.** We split the learning dataset into training, validation, and test data in the ratio of 6:2:2. The training data were used to construct the surrogate model, and the prediction accuracy of the model was evaluated using the validation/test data. The validation data

were specifically used to determine the stopping point of model training (see the "Model training" subsection), whereas the test data were used to assess the performance of the trained model.

## Machine learning

**Deep neural network.** We used a fully connected DNN as a regression model to fit the training data. The DNN comprises a total of $N_{\text{layer}} + 2$ layers: an input layer, a series of $N_{\text{layer}}$ hidden layers, and an output layer (S1 Fig). The input and output layers include nodes equal to the number of inputs and outputs, respectively. Each hidden layer comprises an equal number of nodes, $N_{\text{node}}$, and each node is connected to all nodes in the adjacent layers. Initially, the values of $\{x_n\}_{n=1}^{60}$ serve as input to the nodes in the input layers. Subsequently, each node in the first hidden layer receives the weighted inputs, sums them up, adds a bias, and finally applies the rectified linear unit (ReLU) activation. This process continues for each layer up to the last hidden layer. The nodes in the last hidden layer and the output layer are fully connected without ReLU activation. Consequently, the DNN turns into a recursive function, as follows:

$$\boldsymbol{y}^l = \begin{cases} \boldsymbol{x}, & (l = 1) \\ \max(0, \boldsymbol{W}^l \boldsymbol{y}^{l-1} + \boldsymbol{b}^l), & (2 \leq l \leq N_{\text{layer}} + 1) \\ \boldsymbol{W}^l \boldsymbol{y}^{l-1} + \boldsymbol{b}^l, & (l = N_{\text{layer}} + 2) \end{cases} \tag{11}$$

where $\boldsymbol{y}^l$, $\boldsymbol{W}^l$, and $\boldsymbol{b}^l$ denote the output vector, weight matrix, and bias vector of the $l$-th layer, respectively.

**Model training.** The DNN was trained using the data by adjusting the weights and biases to minimize the loss function, which is defined as the mean squared error of the outputs, as follows:

$$\mathcal{L} = \frac{1}{N_{\text{sample}}} \cdot \frac{1}{N_{\text{out}}} \sum_{s=1}^{N_{\text{sample}}} \left\| \boldsymbol{y}_{\text{sim}}^{(s)} - \boldsymbol{y}_{\text{DNN}}^{(s)} \right\|^2, \tag{12}$$

where $\|\cdot\|$ denotes the Euclidean norm ($l_2$-norm of a vector), $N_{\text{out}} = 45$ indicates the number of outputs, $N_{\text{sample}}$ represents the total number of samples used for evaluation, $\boldsymbol{y}_{\text{sim}}^{(s)}$ denotes the outputs in the training data (outputs from the 1D–0D simulation), and $\boldsymbol{y}_{\text{DNN}}^{(s)}$ indicates the outputs predicted by the DNN. We used the gradient-based algorithm "Adam" [59] for optimization, with an initial learning rate $lr$. Furthermore, mini-batch training was employed with a batch size $N_{\text{batch}}$, and batch normalization [60] was applied between the linear transformation and ReLU activation in individual layers.

During the training, the coefficient of determination ($R^2$ score), defined as

$$R^2 = 1 - \frac{\sum_{s=1}^{N_{\text{sample}}} \left\| \boldsymbol{y}_{\text{sim}}^{(s)} - \boldsymbol{y}_{\text{DNN}}^{(s)} \right\|^2}{\sum_{s=1}^{N_{\text{sample}}} \left\| \boldsymbol{y}_{\text{sim}}^{(s)} - \bar{\boldsymbol{y}}_{\text{sim}} \right\|^2}, \tag{13}$$

was evaluated based on the validation data at the end of each epoch to assess the performance improvements. Herein, $\bar{\boldsymbol{y}}_{\text{sim}}$ denotes the mean of $\boldsymbol{y}_{\text{sim}}^{(s)}$. The closer the $R^2$ score is to 1, the more precise the prediction; $R^2 = 1$ if $\boldsymbol{y}_{\text{DNN}}^{(s)}$ and $\boldsymbol{y}_{\text{sim}}^{(s)}$ are equal for all $s$. For every 100 epochs, we monitored the $R^2$ score averaged over the latest 100 epochs; training was terminated if no improvements were observed in three successive evaluations. The weights and biases with the highest $R^2$ scores at the epoch were selected for the trained model. The DNN and its training were implemented using "Chainer" [61], which is a Python-based open-source framework for deep learning.

Training the DNN involves certain hyperparameters, namely $N_{\text{layer}}$, $N_{\text{node}}$, $N_{\text{batch}}$, and $lr$, which are not trainable through the optimization process as they are chosen arbitrarily. Although the choice of hyperparameters affects the prediction accuracy of the trained model significantly, the optimal values cannot be known in advance as they vary considerably depending on the data. Therefore, we conducted a grid search to identify the best combination of hyperparameters in $N_{\text{layer}} \in \{5, 7, 10, 13\}$, $N_{\text{node}} \in \{50, 100, 200, 400\}$, $N_{\text{batch}} \in \{300, 1000, 3000, 10000\}$, and $lr \in \{10^{-3}, 10^{-2.5}, 10^{-2}, 10^{-1.5}\}$. After $4^4 = 256$ rounds of training, the $R^2$ scores evaluated by the models based on the test data were compared, and the best-performing model was selected as the final surrogate model.

The training data were preprocessed to improve the model performance. We normalized the inputs such that their upper and lower bounds were 1 and $-1$, respectively, and standardized the outputs to ensure zero mean and unit SD. The inputs of the validation and test data were normalized similar to the training data, whereas the outputs were scaled as $y' = (y - \mu_{\text{train}})/\sigma_{\text{train}}$. Herein, $\mu_{\text{train}}$ and $\sigma_{\text{train}}$ denote the mean and SD of the outputs of the training data, respectively. The normalization and standardization (also known as z-score normalization) applied to the data herein constitute standard preprocessing in supervised learning [62].

**Model verification.**  The surrogate model was verified by (i) assessing the prediction accuracy using 40 000 samples of test data, and (ii) comparing the surrogate model and 1D–0D simulation in terms of the predicted outputs and adjusted inputs of the seven patients (Table 1). In the second step, the procedure for assigning or adjusting the inputs based on the patient's clinical data during the prediction performed by the surrogate model was identical to that of the simulation ("Patient-specific modeling" subsection). In both steps, we used the mean absolute error (MAE)

$$MAE_m = \frac{1}{N_{\text{sample}}} \sum_{s=1}^{N_{\text{sample}}} |y^{(s)}_{\text{DNN},m} - y^{(s)}_{\text{sim},m}|, \quad m = 1, 2, \ldots, 45 \tag{14}$$

in addition to the $R^2$ score to assess the prediction accuracy of the surrogate model.

## Uncertainty quantification and sensitivity analysis

We used the surrogate model to perform the UQ while predicting the risk of postoperative CH in three patients with ICA stenosis. This demonstrated the application of the surrogate model to the UQ problem and facilitated the investigation of the relationship between collateral circulation in the CoW and CH.

**Quantity of interest.**  CH is defined as an increase of more than 100% in the flow rate of cerebral arteries due to ICA stenosis surgery [35–38]. Therefore, we focused on predicting the cerebral circulation when the stenosis is dilated, to evaluate the percentage increase in outflows of the CoW as follows:

$$\Delta \bar{Q}_i = \frac{\bar{Q}_i^{\text{post}} - \bar{Q}_i^{\text{pre}}}{\bar{Q}_i^{\text{pre}}} \times 100\%, \quad i = 1, 2, \ldots, 6. \tag{15}$$

Herein, $\bar{Q}_i^{\text{pre}}$ and $\bar{Q}_i^{\text{post}}$ denote the cycle-averaged flow rates at the six outlets of the CoW before and after dilating the stenosis, respectively. By definition, $\Delta \bar{Q}_i > 100\%$ represents CH.

**Target patient characteristics.**  Three patients (Patients 1–3 in Table 1) were included in the surgical outcome prediction. The imaging data (CT), measurements of inflows (PC-MRI or ultrasound) and outflows (SPECT) of the CoW, and mean arterial pressure collected before the surgery were used for the predictions. The evaluated stenosis ratios (*SR*) for Patients 1, 2, and 3 were 59%, 83%, and 91%, and the corresponding $R_v$ values (evaluated using Eq (5)) were 0.5 mmHg s mL$^{-1}$, 11.3 mmHg s mL$^{-1}$, and 66.6 mmHg s mL$^{-1}$, respectively, exhibiting the

same trend as the *SR* (see S2 Fig for the stenosis geometry). Patients 1 and 3 each had a complete CoW; however, CT images of Patient 2 suggested hypoplasia (missing) of an anterior communicating artery (ACoA). Patient 2 was identified by the surgeon as being at risk for CH based on the collected data. To minimize the potential risk of CH, Patient 2 underwent staged surgery, where the stenosis was pre-dilated using a balloon, followed by complete dilation with a stent after two weeks.

**Uncertainty modeling.**   We evaluated the uncertainty in the clinical data that were used to assign or adjust the patient-specific inputs. We focused on uncertainties in the arterial diameters and stenosis parameters, which were used directly as inputs, and those in the CoW inflow and outflow measurements, which were used to obtain the target outputs. The arterial length is more robustly measured than the diameter and has a minor effect on flow resistance; therefore, the uncertainty in length was not considered.

In all three patients, arterial diameters and stenosis parameters were obtained through segmentation of the arterial lumen on CT images. The geometry obtained during the segmentation can vary based on the threshold used to determine the boundary. In the case of CT, the lumen boundary spanned 2–3 pixels, and the diameter changed by ±2 pixels based on the threshold used. Therefore, we assumed uncertainty of ±2 pixels (±0.702–0.936 mm, depending on image resolution) with respect to the arterial diameter obtained from the segmentation. Similarly, uncertainties in the stenosis parameters were estimated by considering a 2-pixel uncertainty in the underlying geometry. However, an exception was made for Patient 2, as the ACoA was not recognized on CT images of this patient, suggesting hypoplasia of the ACoA. Nevertheless, we could not rule out the possibility that the ACoA, hidden between the extremely close presence of the left and right anterior cerebral arteries, might have failed to resolve on the images (Fig 3). Therefore, we assumed uncertainty of 0.1–2.6 mm in the ACoA diameter, thereby including the possibility of its absence as well as presence.

Uncertainties in the measured flow rates were determined based on modality. The uncertainties in the measured values were assumed to be ±16%, ±35%, and ±16% for PC-MRI, ultrasound, and SPECT, respectively, based on the literature [19,63–68] and discussions with surgeons. The uncertainty ranges were intentionally overestimated to maximize the chance of including the "true" (yet unknown) value regardless of the modality used.

**Uncertainty propagation.**   Uncertain inputs and targets were treated as random variables with uniform distribution on the determined interval. To estimate the statistics of the predicted $\Delta \bar{Q}_i$ under uncertainties, we used the most straightforward approach for UQ, namely the Monte Carlo method. In each realization, uncertain inputs and targets were sampled from

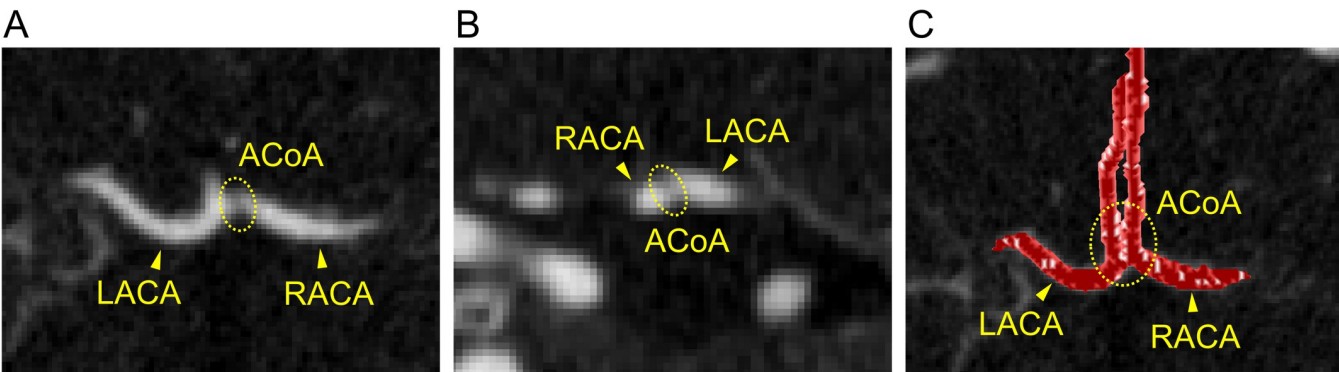

**Fig 3. Computed tomography (CT) images of Patient 2.** (A) Transverse plane, (B) frontal plane, and (C) volume-rendered image. The ACoA was not recognized on CT images of this patient, suggesting hypoplasia of the ACoA.
ACoA, anterior communicating artery; LACA, left anterior cerebral artery; RACA, right anterior cerebral artery.

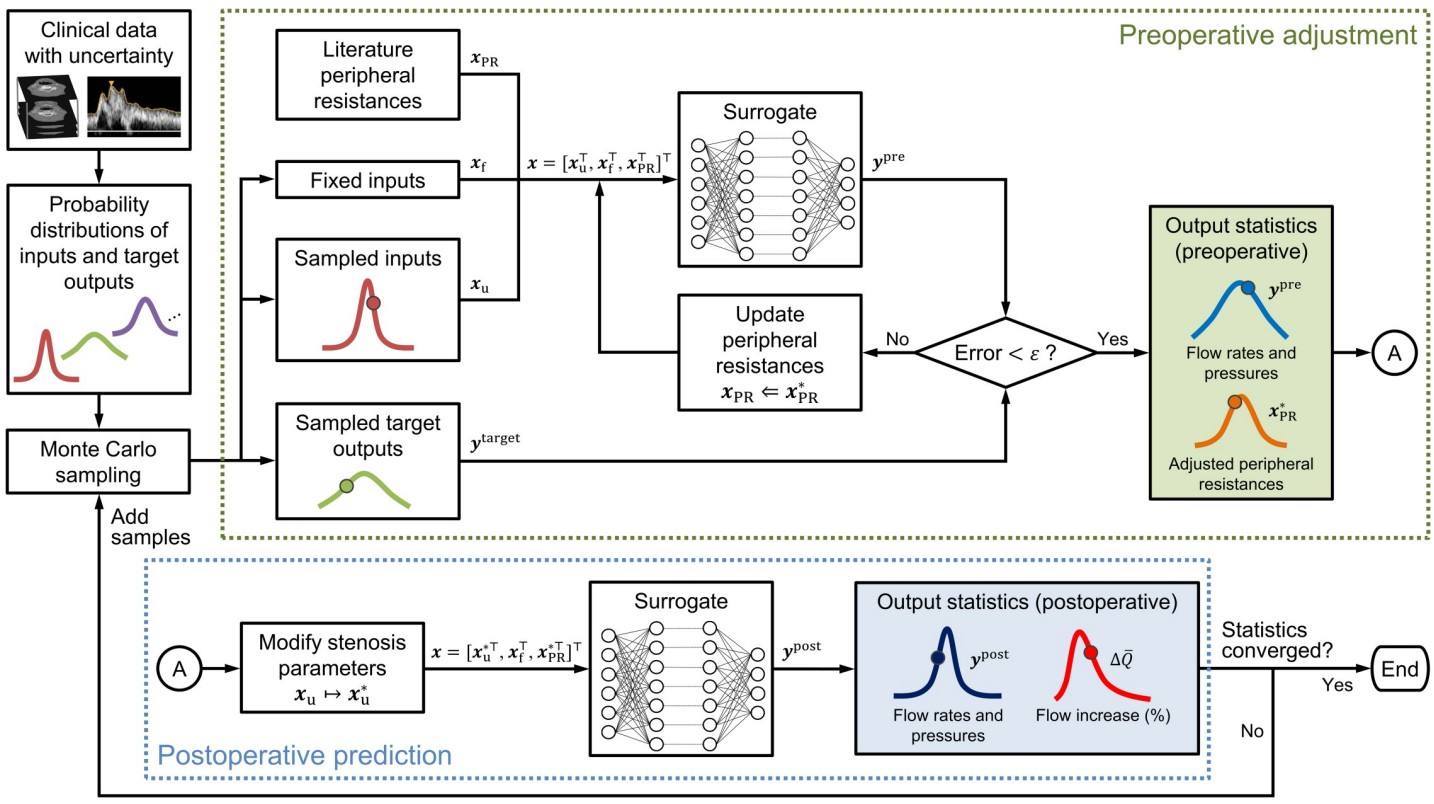

**Fig 4. Flowchart for uncertainty quantification using the Monte Carlo method.** For each Monte Carlo sample, peripheral resistances of the circle of Willis and the scaling factor for total peripheral resistance were adjusted ("preoperative adjustment"), followed by a virtual dilation of the stenosis to predict the cerebral circulation immediately after the surgery ("postoperative prediction"). The number of samples was increased sequentially until the statistics converged. The method can be applied to any probability density function; however, we assume a uniform distribution in this study. Additional details regarding the algorithm are provided in S2 Appendix.

a specified probability distribution, and $\Delta \bar{Q}_i$ was predicted through successive steps of "preoperative adjustment" and "postoperative prediction" (Fig 4). In the first step, the PRs of the CoW and scaling factor for the total PR were adjusted to match the predicted outputs to the targets. The samples were rejected if target convergence was not attained. Subsequently, the stenosis parameters were modified to $R_v = 0$, $D_n = D_{ICA}$, $SR = 0$, and $K_t = 0$ to reflect the complete dilation of the stenosis. The modified stenosis parameters and adjusted PRs were used as inputs in the subsequent steps to predict cerebral circulation immediately after the stenosis surgery. Finally, $\Delta \bar{Q}_i$ was calculated using the flow rates before and after the surgery. The statistics of $\Delta \bar{Q}_i$ under uncertainties were estimated using the collected $\{\Delta \bar{Q}_i^{(s)}\}_{s=1}^{N_{MC}}$, where $N_{MC}$ denotes the number of realizations. $N_{MC}$ was increased sequentially until the statistics of $\Delta \bar{Q}_i$ converged. As a basic policy, we increased $N_{MC}$ by 10 000 and ensured that the change in mean and variance of $\Delta \bar{Q}_i$ was within 0.1%. We also confirmed that there was no significant change in the probability of $\Delta \bar{Q}_i > 100\%$ when $N_{MC}$ was increased. A detailed description of the algorithm for uncertainty propagation is provided in S2 Appendix.

In this study, we assumed that the surgery did not alter the arterial geometry (except for stenosis) and PRs. This assumption is justified because we aim to predict the cerebral circulation immediately after the surgery. Additionally, autoregulation and remodeling of the cerebral arteries generally prevent an abrupt change in blood flow. Therefore, our assumption is appropriate for predicting the maximum possible $\Delta \bar{Q}_i$, which is the most dangerous surgical outcome in terms of CH.

**Sensitivity analysis.**   In addition to UQ, we performed SA to measure the impact of each parameter (uncertain input or target) on $\Delta\bar{Q}_i$. We adopted a variance-based global SA proposed by Sobol' [69] to consider the interaction between the parameters. In this method, the impact of parameter $x_n$ on output $y$ is quantified as the Sobol' sensitivity indices [69,70]:

$$S_n = \frac{\mathbb{V}[\mathbb{E}[y|x_n]]}{\mathbb{V}[y]}, \tag{16}$$

$$S_{\mathrm{T},n} = 1 - \frac{\mathbb{V}[\mathbb{E}[y|\boldsymbol{x}_{-n}]]}{\mathbb{V}[y]}, \tag{17}$$

where $\mathbb{E}[y|x_n]$ denotes the conditional expectation of $y$ for a fixed $x_n$; $\mathbb{V}[y]$ indicates the variance of $y$; $\boldsymbol{x}_{-n}$ represents all parameters except $x_n$; $S_n$, the first-order sensitivity index, quantifies the independent contribution of $x_n$ to the measured variability of $y$; and $S_{\mathrm{T},n}$, the total sensitivity index, quantifies the overall contribution of $x_n$ to the variability of $y$, including indirect contributions through interactions with other parameters. A large $S_{\mathrm{T},n} - S_n$ indicates that the impact of $x_n$ varies significantly with the values of other parameters.

We used Saltelli's algorithm with the Monte Carlo method to compute the sensitivity indices [70, 71]. The accuracy of the sensitivity indices in terms of the sampling error was assessed by estimating the 95% confidence interval using the bootstrap method [72] with a sample size of 1000. The SA was implemented using the open-source Python library "SALib" [73].

## Results

### Surrogate modeling

**Effect of hyperparameters.**   Based on the grid search for 256 sets of hyperparameters, the highest $R^2$ score was achieved when $N_{\mathrm{layer}} = 7$, $N_{\mathrm{node}} = 200$, $N_{\mathrm{batch}} = 3000$, and $lr = 10^{-2.5}$ (S3 Fig). Combinations with $N_{\mathrm{node}} = 200$ yielded an overall higher $R^2$ score, indicating that $N_{\mathrm{node}}$ affects the $R^2$ score more than the other hyperparameters.

$N_{\mathrm{layer}}$ and $N_{\mathrm{node}}$ determine the total number of trainable parameters (weights and biases) of the DNN, whereas $N_{\mathrm{batch}}$ and $lr$ control the gradient and the rate of parameter update, respectively, during the optimization process. To compare the influences of these effects on prediction accuracy, we plotted the $R^2$ score with respect to the number of trainable parameters, as illustrated in Fig 5A, using the following equation:

$$N_{\mathrm{param}} = \sum_{l=2}^{N_{\mathrm{layer}}+2}(N_{\mathrm{node}}^l \cdot N_{\mathrm{node}}^{l-1} + N_{\mathrm{node}}^l), \tag{18}$$

where $N_{\mathrm{node}}^l$ denotes the number of nodes in the $l$-th layer. Note that the index of summation starts at 2 instead of 1 since the input layer ($l = 1$) has no parameters. Fig 5A indicates that the $R^2$ score has an inverted U-shaped relationship with $N_{\mathrm{param}}$. The highest $R^2$ score was achieved when the DNN contained 262 445 trainable parameters; increasing or decreasing the number of parameters from this optimal number resulted in lower $R^2$ scores. The vertical variations in the $R^2$ score indicate that the effects of $N_{\mathrm{batch}}$ and $lr$ were relatively small when the DNN comprised an optimal number of parameters. However, the choice of $N_{\mathrm{batch}}$ and $lr$ significantly affected the prediction accuracy when training the DNN with more than 1 million parameters.

**Effect of number of training samples.**   To investigate the effect of the number of training samples on the $R^2$ score, we trained the DNN with different numbers of training samples while maintaining the hyperparameters constant in the optimal combination. Fig 5B compares the networks' $R^2$ scores evaluated using identical test data. Increasing the number of training samples improved the prediction accuracy significantly, particularly with a smaller number of

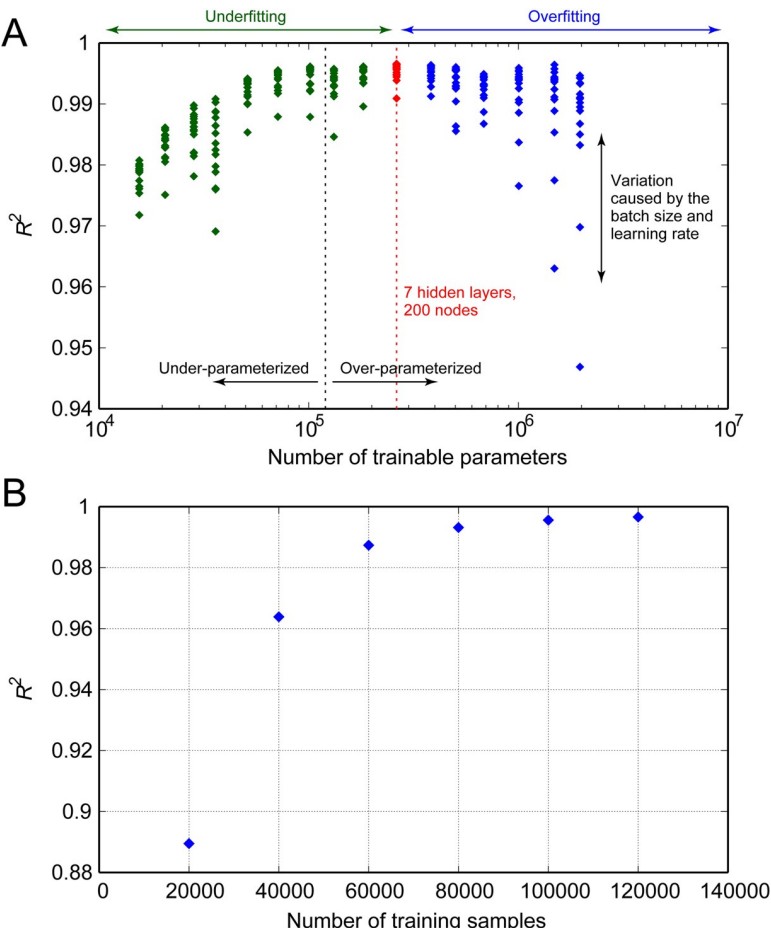

**Fig 5. Changes in the $R^2$ score of the trained model.** (A) Changes with respect to the number of trainable parameters in the deep neural networks. The number of training samples was maintained constant at 120 000, and the $R^2$ scores were evaluated using 40 000 test samples. Under- or over-parameterized indicate that the networks contain fewer or more trainable parameters than the number of training data, respectively. (B) Changes in the $R^2$ score with respect to the number of samples used for training.

samples. However, the accuracy reached a plateau with 120 000 samples, indicating that the accuracy cannot be improved further with more samples.

**Model performance.** The best-performing DNN, trained with hyperparameters $N_{layer} = 7$, $N_{node} = 200$, $N_{batch} = 3000$, and $lr = 10^{-2.5}$, was selected as the final surrogate model and verified. Initially, we assessed the prediction accuracy of the model using 40 000 samples of test data. The overall $R^2$ scores for the flow rate and pressure were 0.9959 and 0.9973, respectively. On average, the MAE was 2.617 mL/min for the flow rate and 0.7226 mmHg for the pressure, which correspond to approximately 4% and 0.9% of the flow rate and pressure mean absolute values, respectively. A detailed comparison of the flow rate and pressure in each artery predicted by the surrogate model and 1D–0D simulation are illustrated in S4 and S5 Figs.

Furthermore, to verify the model accuracy using the patients' clinical data for assigning and adjusting the inputs, we compared the surrogate model and 1D–0D simulation in terms of flow rates, pressures, and adjusted PRs of the CoW for the seven patients (Fig 6). The flow rates at the six outlets of the CoW were excluded from the evaluation, as they matched the measured flow rates. As indicated in Fig 6, the outputs and adjusted PRs from the surrogate model were in agreement with those from the simulation. Even in the case of patient-specific

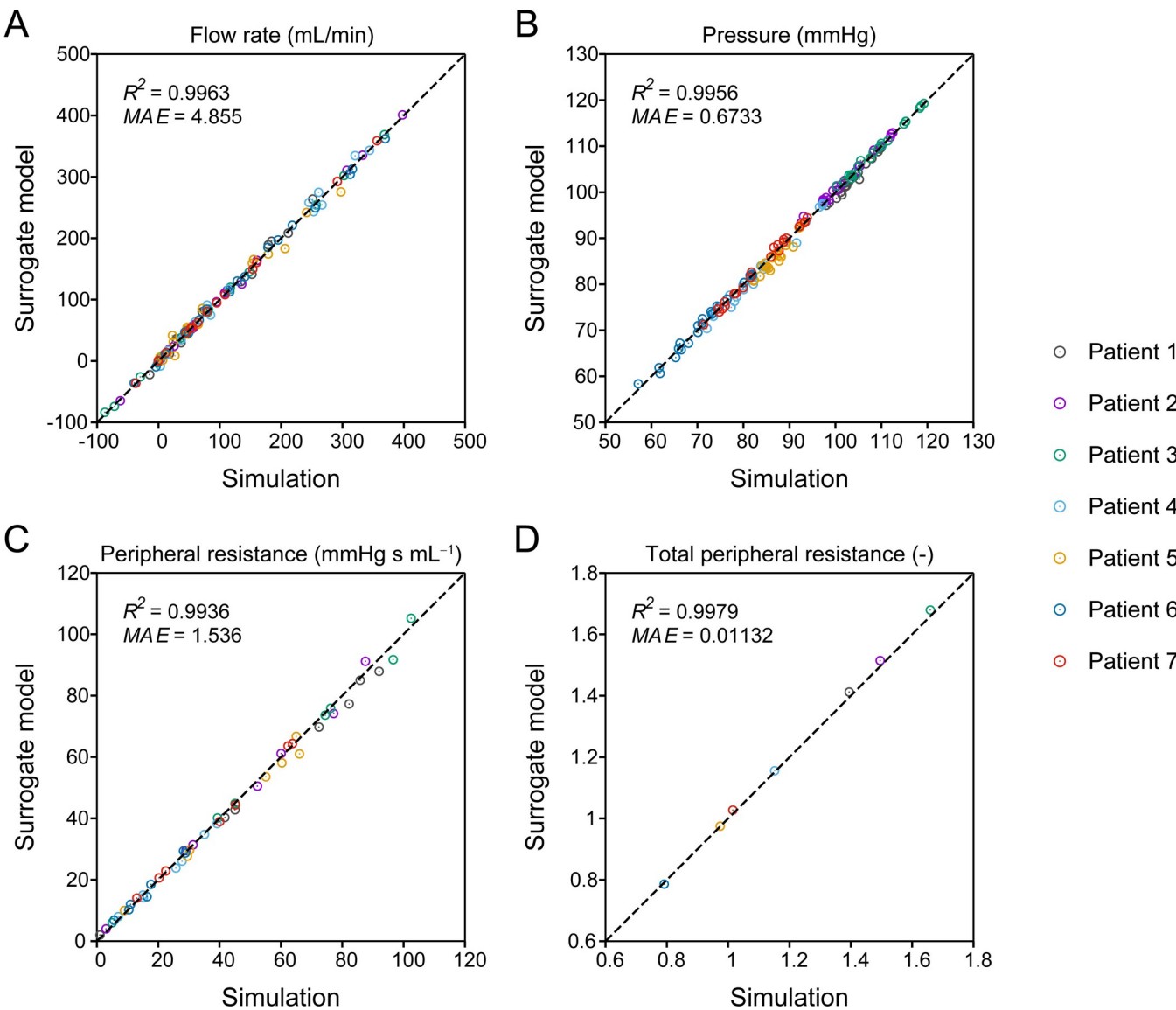

**Fig 6. Comparison of one-dimensional–zero-dimensional (1D–0D) simulation and surrogate model predictions.** The (A) flow rate, (B) pressure, (C) adjusted peripheral resistance of the circle of Willis, and (D) adjusted scaling factor for total peripheral resistance in seven patient-specific cases are compared. The negative flow rate indicates that the flow direction is opposite to the arrows in Fig 2. The $R^2$ scores and mean absolute errors (MAEs) of each quantity are depicted in the corresponding panels.

predictions that involved iterative adjustment of inputs, the flow rate and pressure errors were comparable to those evaluated using the test data.

Fig 7 compares the surrogate model and simulation in terms of the time required for a single prediction. On a single CPU core (Intel Core i9-9900K, 3.6 GHz), the surrogate model achieved a prediction time of several milliseconds, reducing the computation time of the simulation by a factor of over 43 000. Furthermore, the surrogate model exhibited excellent parallelization performance, particularly when executed on a GPU (NVIDIA GeForce RTX2080 Ti), and significantly reduced the computation time per prediction. As illustrated in Fig 7, the computation time was only five times longer when the surrogate model performed 10 000 predictions on a GPU than a single prediction on a single CPU core. Parallelization on the GPU

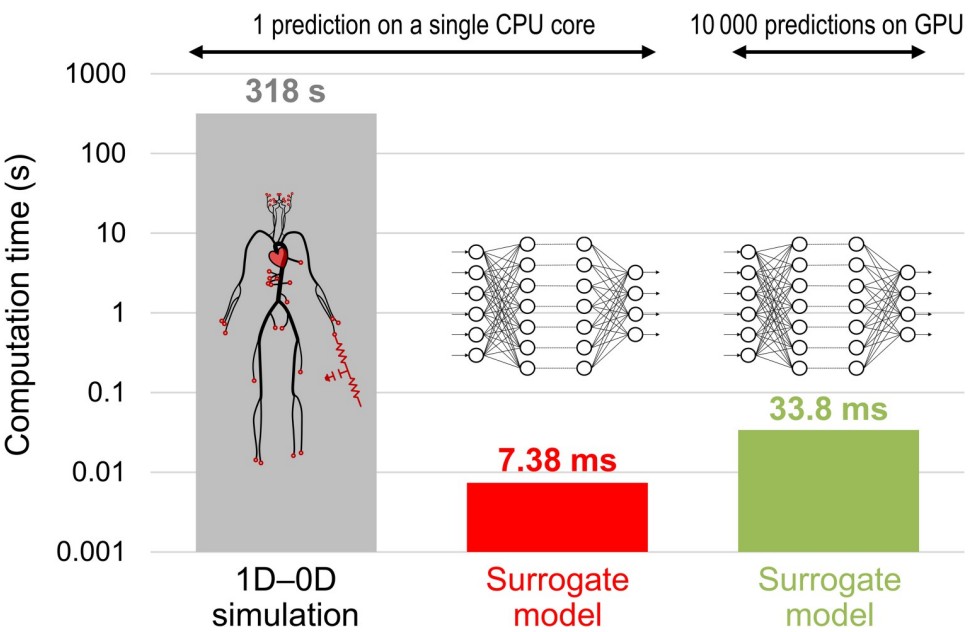

**Fig 7. Comparison of the time required for prediction.** Computation times for a one-dimensional–zero-dimensional (1D–0D) simulation and surrogate model on one CPU core and a surrogate model on GPU are compared.

was performed using the built-in backend of Chainer [61] for CUDA-based parallel matrix operations. The latest deep learning libraries, including TensorFlow, Keras, PyTorch, and Chainer, support GPU execution using their built-in backends, allowing easy parallelization of matrix operations in training and predictions.

### Uncertainty quantification and sensitivity analysis

**Flow rate increase following the stenosis surgery.** The percentage increase in flow rate ($\Delta \bar{Q}$) following the ICA stenosis surgery was evaluated for Patients 1–3, considering the uncertainties in arterial diameters, stenosis parameters, and target flow rates. The number of realizations ($N_{\mathrm{MC}}$) to obtain the statistics of $\Delta \bar{Q}$ was set to 100 000. The time required for the UQ was a few minutes on a single CPU core, which was shorter than the time required for a single prediction using the 1D–0D simulation. The time was reduced to less than a minute when executed on a GPU.

Although we obtained $\Delta \bar{Q}$ at each outlet of the CoW, we focused only on the results at the middle cerebral artery (MCA) on the stenosis side, which was subjected to the largest $\Delta \bar{Q}$. Fig 8 depicts the probability density of the predicted $\Delta \bar{Q}$ for Patients 1–3 along with the values from the deterministic 1D–0D simulation (represented as triangles). Additionally, the figure depicts the interval, mean, and mode (the value with the highest frequency) of $\Delta \bar{Q}$ and the probability of $\Delta \bar{Q}$ being more than 100%. A negative $\Delta \bar{Q}$ indicates a decrease in the flow rate following the surgery. This situation may be rare in actual patients with severe stenosis; nonetheless, it is not non-physiological, as observed in certain clinical cases [36].

Overall, uncertainties in the clinical data generated large variations in the predicted $\Delta \bar{Q}$. Based on the comparison of patients' results, we observed that the mode of $\Delta \bar{Q}$ was close to the $\Delta \bar{Q}$ predicted by the deterministic simulation and higher when stenosis was more severe (larger $SR$ and $R_v$). In all patients, the distribution of $\Delta \bar{Q}$ was skewed to the right, with a higher mean than the mode. The distribution of $\Delta \bar{Q}$ was spread extensively to large values in Patients

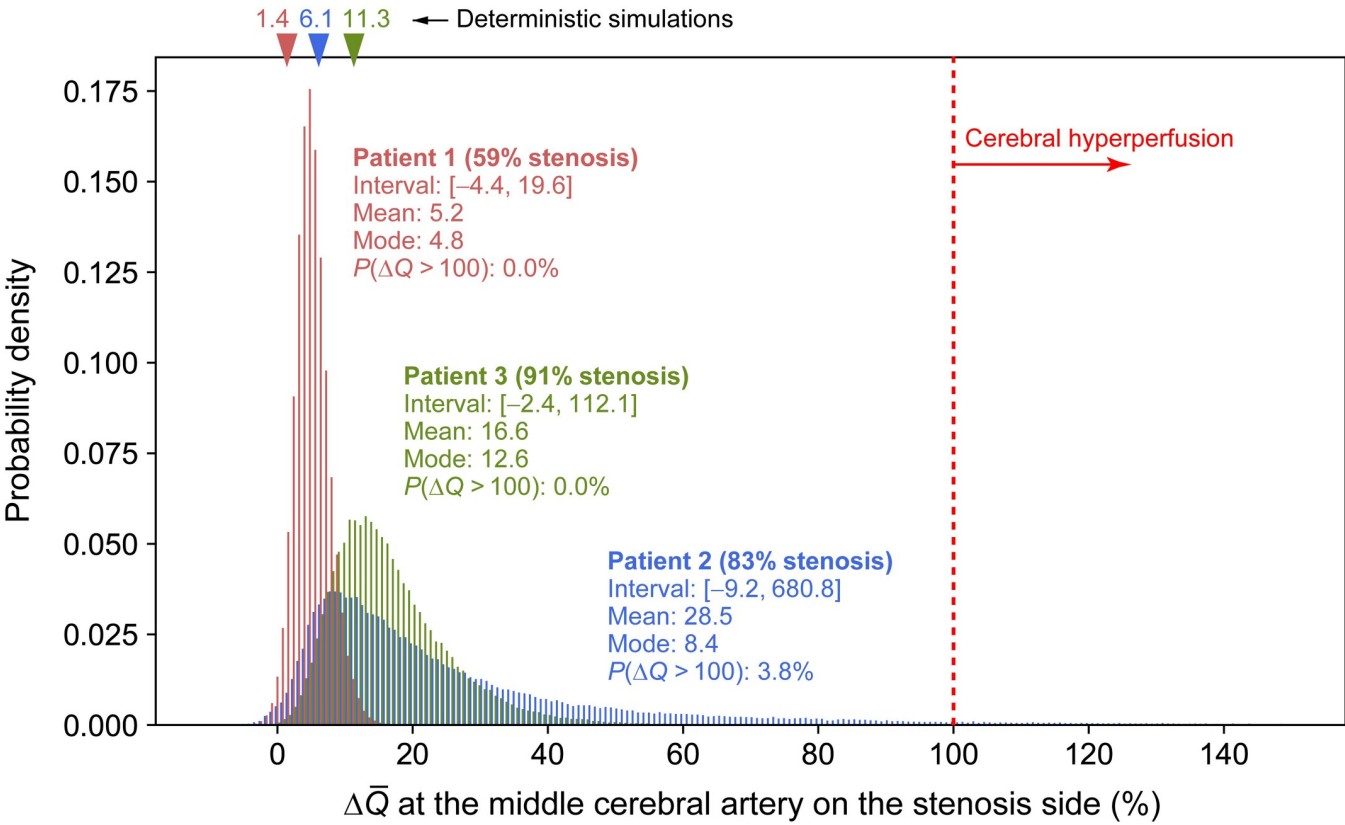

**Fig 8. Probability density of the predicted value of postoperative flow increase ($\Delta\bar{Q}$).** Flow increase at the middle cerebral artery on the stenosis side is illustrated for Patients 1–3. Triangles indicate the values predicted by one-dimensional–zero-dimensional (1D–0D) simulation without considering uncertainties.

2 and 3, wherein the stenosis was more severe than in Patient 1. The increase in the prediction uncertainty in $\Delta\bar{Q}$ with higher stenosis severity is attributed to the 2-pixel uncertainty considered for the arterial diameter. With the same variation width of diameter, the uncertainty in $R_v$ (Eq (5)) and $SR$ ($= 1 - D_s/D_n$) increases with a smaller diameter, leading to a larger uncertainty in $\Delta\bar{Q}$.

However, the comparison of Patients 2 and 3 indicated that CH ($\Delta\bar{Q} > 100\%$) is not caused solely by the severity of stenosis. Patient 2 exhibited a 3.8% chance of CH, whereas the corresponding estimates for Patients 1 and 3 were 0% and 0.001% (only one sample out of 100 000 samples), respectively. In Patient 2, who was assumed to have a possible missing ACoA, the variability of $\Delta\bar{Q}$ to values above 100% was prominent compared to Patient 3, implying that $\Delta\bar{Q}$ was significantly affected by this artery.

**Patient conditions causing cerebral hyperperfusion.** To clarify the conditions under which CH occurs, we further investigated the characteristics of 3796 samples in Patient 2 and 1 sample in Patient 3 with $\Delta\bar{Q} > 100\%$. The left column in Fig 9 depicts the relationship between the preoperative PR of the MCA on the stenosis side ($PR_{MCA}$) and $\Delta\bar{Q}$ in each patient. Furthermore, the right column in Fig 9 illustrates the variation in $\Delta\bar{Q}$ with respect to the diameters of the ACoA and the posterior communicating artery (PCoA) on the stenosis side in each patient.

As indicated in the left column of Fig 9, $\Delta\bar{Q}$ exhibits an inverse relationship with $PR_{MCA}$. This is natural because $\Delta\bar{Q} \propto \Delta\bar{P}_{MCA}/PR_{MCA}$, where $\Delta\bar{P}_{MCA}$ denotes pressure recovery at the

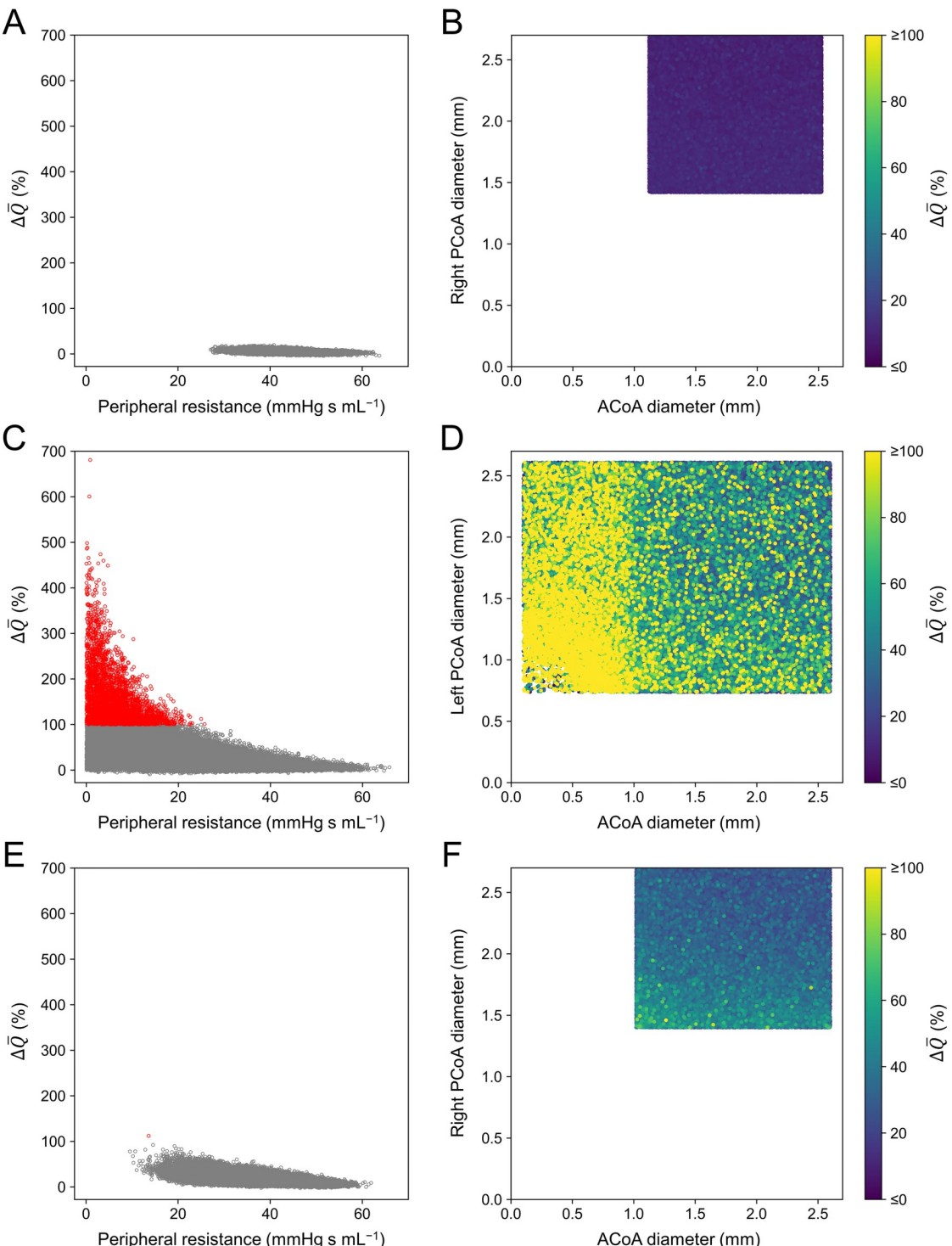

**Fig 9. Postoperative flow increase ($\Delta \bar{Q}$) in Patients 1–3 relative to several factors.** Left column: scatter plot of $\Delta \bar{Q}$ at the middle cerebral artery on the stenosis side with respect to the adjusted preoperative peripheral resistance of this artery. Samples with $\Delta \bar{Q} > 100\%$ are indicated in red. Right column: $\Delta \bar{Q}$ with respect to the diameters of the anterior communicating artery (ACoA) and posterior communicating artery (PCoA) that form the collateral pathway to the artery on the stenosis side. Samples with $\Delta \bar{Q} > 100\%$ are depicted in yellow, regardless of their value. (A) (B) Patient 1; (C) (D) Patient 2; and (E) (F) Patient 3.

MCA on the stenosis side caused by the surgery. Even with the same $\Delta\bar{P}_{\mathrm{MCA}}$, a smaller $PR_{\mathrm{MCA}}$ results in a larger $\Delta\bar{Q}$. Fig 9C and 9E depict the results of Patients 2 and 3, respectively, wherein the $PR_{\mathrm{MCA}}$ is smaller than 20 mmHg s mL$^{-1}$ in most samples when $\Delta\bar{Q}$ exceeds 100%. However, we observed that a small $PR_{\mathrm{MCA}}$ did not always result in $\Delta\bar{Q} > 100\%$, as $\Delta\bar{P}_{\mathrm{MCA}}$ varied with respect to some factors. Samples with $\Delta\bar{Q} > 100\%$ were associated not only with a small $PR_{\mathrm{MCA}}$ but also with small diameters of the ACoA and PCoA that form the collateral pathway to the artery on the stenosis side (Fig 9D and 9F). Particularly, an extremely small ACoA diameter (<1 mm) resulted in $\Delta\bar{Q} > 100\%$, regardless of the PCoA diameter (Fig 9D).

Fig 10 depicts the variation in the preoperative flow rate in the ACoA (left column) and PCoA (right column) with respect to the diameter. Note that the flow rate shown in Fig 10 varies both horizontally and vertically. As indicated by the relationship between $\Delta P$ and $R_v Q$ in Eq (4), the flow rate in an artery is proportional to the pressure difference between the two ends and is inversely proportional to the fourth power of the diameter. The horizontal variation in flow rate shown in Fig 10 is attributed to the diameter variation of the communicating artery within the uncertainty range. On the contrary, the vertical variation is caused by variations in the pressure difference between the ends (i.e., the pressure difference between arteries on the normal and stenosis sides) resulting from uncertainties in the diameter of other arteries, stenosis severity, and flow measurements. As seen from the large vertical variations, the flow rate of the communicating artery is strongly influenced not only by the diameter uncertainty of this artery but also by other uncertainties.

As indicated in Fig 10C, the flow rate in the ACoA of Patient 2 is distributed up to 250 mL/min regardless of the diameter when it is $\geq 1$ mm. However, when the diameter < 1 mm, the flow rate decreases rapidly, and samples with $\Delta\bar{Q} > 100\%$ appear frequently near the upper end of the distribution. In other words, $\Delta\bar{Q}$ exceeds 100% when the collateral flow through the ACoA is limited owing to the small diameter despite the large pressure difference between arteries on the normal and stenosis sides. Conversely, no such condition was observed in Patients 1 and 3. The small amount of collateral flow in Patient 1 indicates that the pressure difference was small, and in Patient 3, the diameters of the ACoA and PCoA were sufficiently large.

**Sensitivity of uncertain parameters.** To gain further insight into the factors associated with CH, we quantified the influence of each uncertain parameter on $\Delta\bar{Q}$ through SA. According to Saltelli's algorithm, the number of samples required to compute the sensitivity indices was 370 000. Fig 11 depicts the first-order ($S_n$) and total ($S_{\mathrm{T},n}$) sensitivity indices. In most parameters, $S_{\mathrm{T},n}$ is considerably larger than $S_n$, indicating a strong interaction between the parameters. In all patients, the diameters of the ACoA and PCoA on the stenosis side influenced the $\Delta\bar{Q}$ significantly. Additionally, the diameters of the anterior cerebral artery I (ACA I) and posterior cerebral artery I (PCA I) exhibited substantial sensitivity. As depicted in Fig 12, these arteries form collateral pathways to supply blood to the MCA on the stenosis side.

Furthermore, the severity of the stenosis affected the $\Delta\bar{Q}$ considerably. Although both $R_v$ and $SR$ are measures of stenosis severity, only $R_v$ contributed to the variance of $\Delta\bar{Q}$. The impact of $R_v$ on $\Delta\bar{Q}$ was smaller in patients with highly severe stenosis. In Patient 1, the sensitivity of $R_v$ was higher than that of collateral pathway diameters, whereas the opposite behavior was observed in Patients 2 and 3.

## Discussion

### Surrogate modeling approach for uncertainty quantification

Quantifying the impact of uncertainties in clinical data on predictive results is essential for enabling the clinical application of hemodynamic simulations. However, as this task is time-

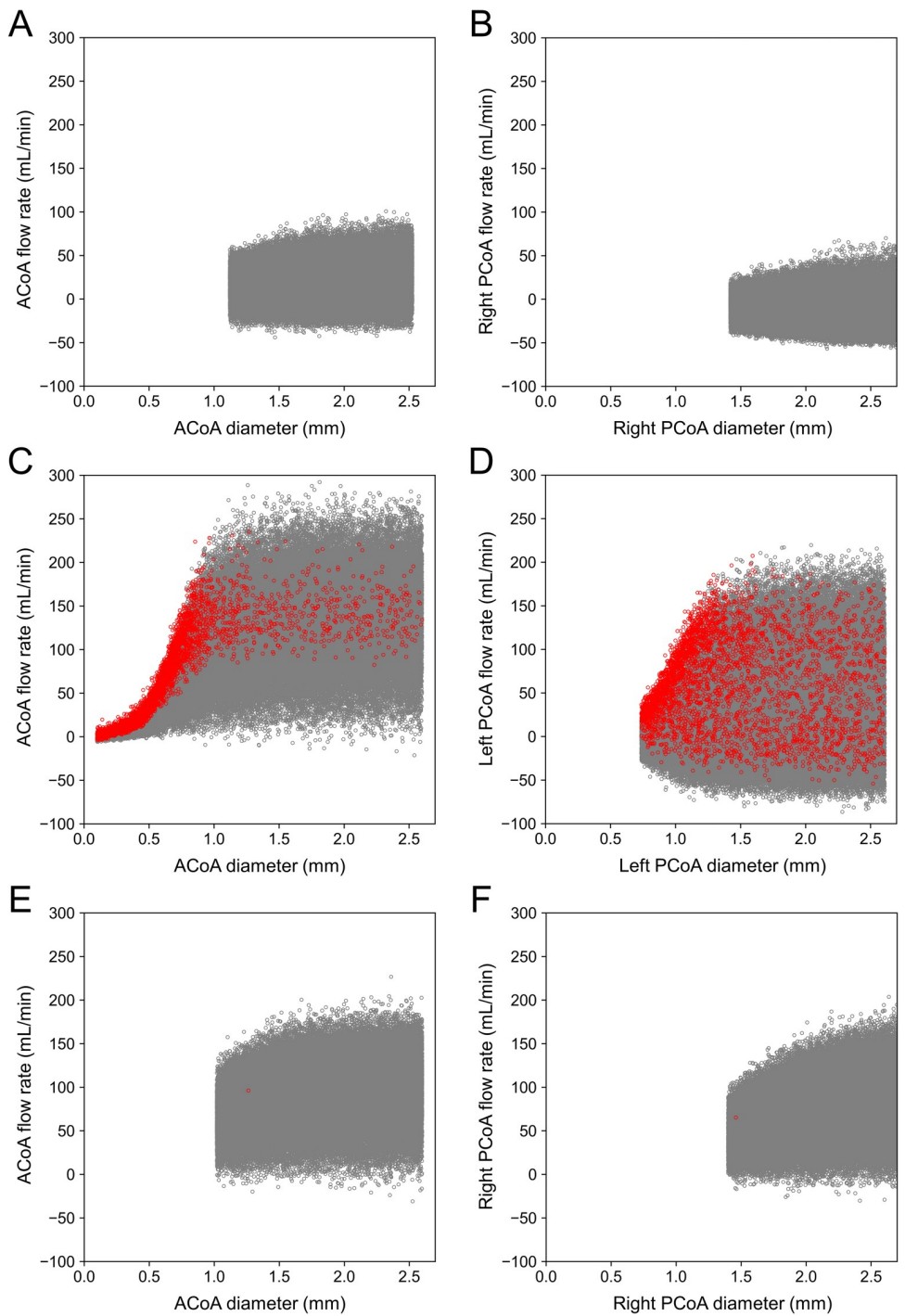

**Fig 10. Scatter plots of preoperative flow rate versus diameter of the communicating arteries in Patients 1–3.** The results for the anterior communicating artery (ACoA) and posterior communicating artery (PCoA) that form the collateral pathway to the artery on the stenosis side are illustrated. The flow rate is indicated as a positive value if blood flows from the artery on the normal side to that on the stenosis side. Samples with $\Delta \bar{Q} > 100\%$ are represented in red. (A) (B) Patient 1; (C) (D) Patient 2; and (E) (F) Patient 3.

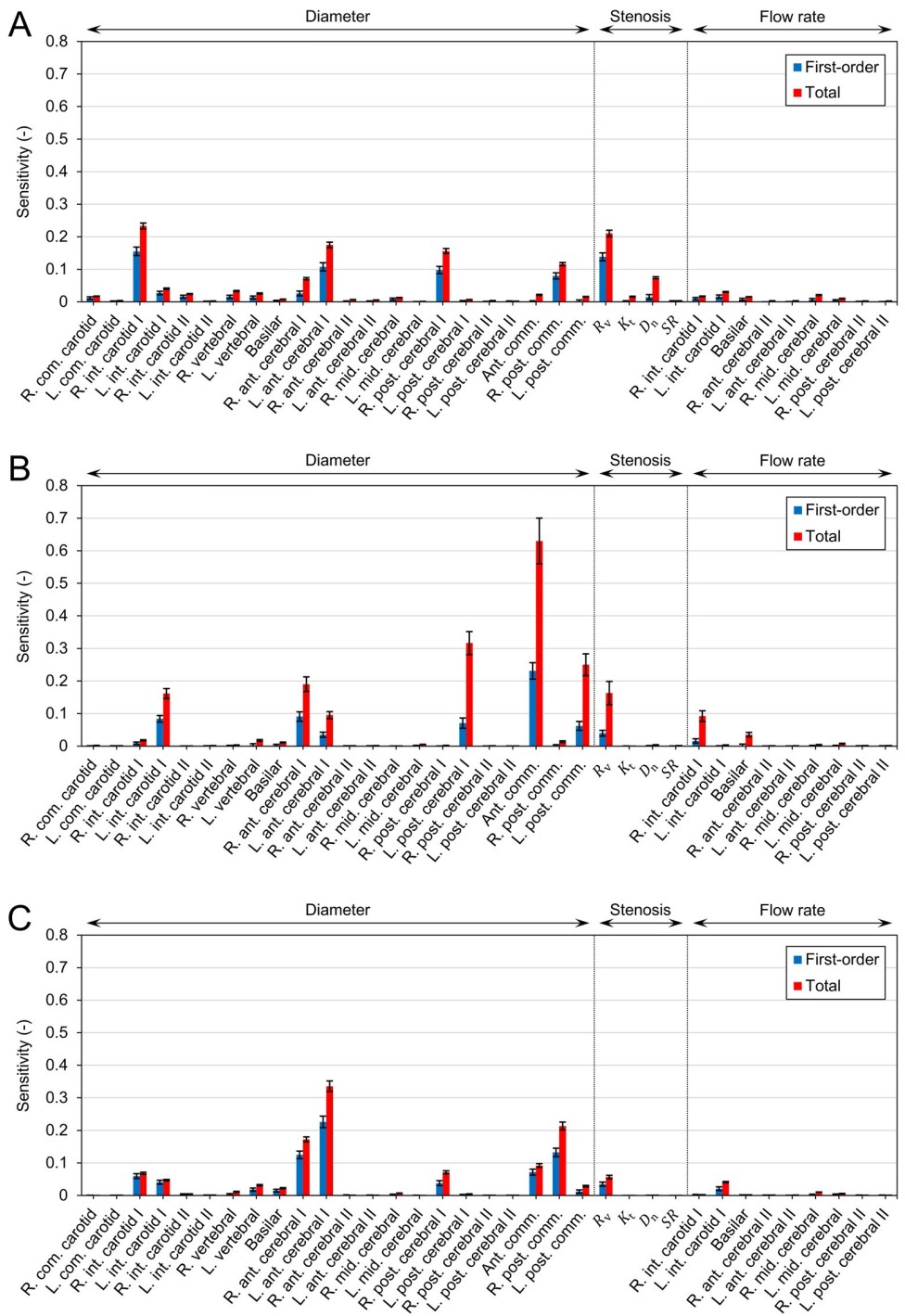

**Fig 11. Sensitivities of uncertain parameters to the postoperative flow increase ($\Delta \bar{Q}$).** The first-order ($S_n$) and total ($S_{T,n}$) sensitivity indices are depicted as bars, and their 95% confidence intervals are represented by black lines. (A) Patient 1, (B) Patient 2, and (C) Patient 3.

consuming and computationally expensive, it is impractical for time-sensitive clinical applications. To address this problem, we trained a DNN using datasets obtained from the 1D–0D simulation to construct a surrogate model that rapidly predicts cerebral circulation subjected

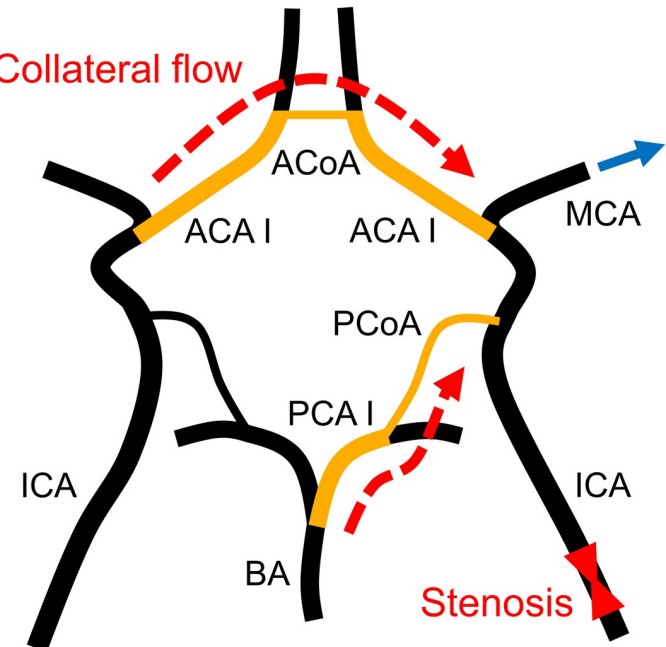

**Fig 12. Collateral flow to the middle cerebral artery downstream of the stenosis.** The middle cerebral artery receives blood supply from the contralateral and posterior inlets to compensate for the reduced blood flow caused by stenosis.

ACA I, anterior cerebral artery I; ACoA, anterior communicating artery; BA, basilar artery; ICA, internal carotid artery; MCA, middle cerebral artery; PCA I, posterior cerebral artery I; PCoA, posterior communicating artery.

to specified geometric and physiological parameters. The DNN predicts the output by computing the input–output relationship, expressed as simple matrix-vector products (Eq (11)), rather than integrating the governing equations through many small time steps to obtain a converged solution. Consequently, the surrogate model reduces the prediction time by a factor of approximately 43 000 in comparison with that of the simulation. In other words, flow rates and pressures in the carotid and cerebral arteries are evaluated in milliseconds (Fig 7). Moreover, as running multiple predictions in parallel only increases the array dimension by one, the surrogate model exhibits excellent parallelization performance. As demonstrated, UQ with 100 000 predictions can be executed nearly in real-time even on a desktop computer, requiring only a few minutes on a single CPU core and less than a minute when using a GPU. The proposed surrogate model facilitates the execution of the existing cost-prohibitive UQ, enabling fast feedback of robust results to the clinic.

During the DNN training, it was evident that the choice of hyperparameters affected the prediction accuracy. For instance, a DNN trained with 120 000 training samples can be less accurate than that trained with only 40 000 samples with respect to the values of the hyperparameters used (Fig 5). The results of the grid search verified that model complexity, which can be represented by the total number of trainable parameters, is a major factor that affects the accuracy of the trained DNN. If the model is extremely simple, it is not sufficiently flexible to represent complex input–output relationships (underfitting). Conversely, if the model is highly complex, the model lacks generalization performance despite well-fitting the training data, resulting in low accuracy on the test data (overfitting). Additionally, the batch size and initial learning rate influence the prediction accuracy; however, the effect is not significant as long as the numbers of hidden layers and nodes are chosen to ensure optimal model complexity.

Furthermore, the number of training samples is important for improving the accuracy of a DNN. The results illustrated in Fig 5B indicate that the performance of machine learning cannot be exploited completely unless sufficient training samples are utilized. In this context, using 1D–0D simulation can effectively obtain a large amount of data in a unified manner with a low computational cost. Although the key to the success of machine learning lies in large datasets, machine learning is an efficient means for surrogate modeling with a limited number of training samples. In this study, training the DNN required 120 000 samples, which was approximately 0.5 times the number of parameters to be determined. This is substantially less than the number of samples required in polynomial chaos, which is a popular approach for surrogate modeling that typically requires oversampling by a factor of 1.5–3 [74]. We also observed that even the DNN trained with as few as 40 000 samples exhibited high performance of $R^2 > 0.96$.

The proposed surrogate model is capable of accurately predicting flow rates and pressures. The predicted outputs are in agreement with those obtained from the 1D–0D simulation for the test data (S4 and S5 Figs) and the patient-specific cases (Fig 6). Particularly, no significant variation was observed in terms of error in cases with different conditions, such as stenosis site and severity, validating that the model can be applied to various patient conditions. This can be attributed to the two approaches used to generate the training data. First, the data were sampled in extremely wide input space. It comprised the entire range that each input could exhibit in an actual patient, considering the inter-patient variability reported in the literature and found in the available clinical data and anatomical variations (S1 Appendix). Second, we considered four possible conditions, with and without ICA stenosis, and obtained sufficient samples for each condition. These approaches ensured that the surrogate model predicted using inputs that were always within the trained input space, avoiding any extrapolation that could significantly decrease accuracy [53].

## Importance of considering clinical data uncertainties

Patient-specific simulations of cerebral circulation use medical images and measurement data to set the geometric and physiological parameters that are appropriate to the patient's condition. However, owing to the limitations of existing imaging technologies, it is difficult to evaluate the diameter of small arteries accurately. For instance, severe stenosis has a diameter less than the spatial resolution of CT scans (approximately 0.4 mm), which implies that a stenosis >90% cannot be evaluated using the images. Similarly, for an ACoA with a diameter of approximately 1.5 mm, a 1-pixel error in lumen segmentation results in a 27% change in diameter. As the flow resistance in a tube is inversely proportional to the fourth power of the diameter (Eq (5)), the effects of a 27% error are significant. This indicates that the UQ facilitated by the proposed surrogate model is necessary to perform reliable predictions.

In this study, we focused on uncertainties in the arterial diameters, stenosis parameters, and flow measurements derived from clinical data and quantified their impact on the predicted value of flow rate increase ($\Delta \bar{Q}$) resulting from the ICA stenosis surgery. The UQ results for the three patients verified that the predicted $\Delta \bar{Q}$ significantly varies with uncertainty (Fig 8). Particularly, the deterministic simulation predicted a higher $\Delta \bar{Q}$ in Patient 3 than in Patient 2 in proportion to the stenosis severity; however, this was reversed in the UQ results, wherein the mean value of $\Delta \bar{Q}$ under uncertainty was higher in Patient 2 than in Patient 3. This validates that predictions that do not consider uncertainties provide only fragmented information, resulting in an inaccurate risk assessment in diagnosis. Moreover, the UQ revealed that Patient 2 had a 3.8% chance of $\Delta \bar{Q}$ being more than 100%, indicating a risk of CH. As predicted by the simulation, Patient 2 was identified by the surgeon as having a risk of CH and

underwent staged surgery to lower the risk. The consistency between the predicted results and the surgeon's judgment further emphasizes the importance of performing UQ and confirms the validity of the proposed approach as a diagnostic tool.

## Biological implications: Cerebral hyperperfusion and collateral circulation

The CoW has a unique ring-like network, wherein the flow from the three inlets is redistributed to the six outlets via communicating arteries. Owing to this network, the MCA on the stenosis side receives collateral flow from the contralateral and posterior inlets through the ACA I, ACoA, PCA I, and PCoA, as depicted in Fig 12. Clinical studies report that CH after ICA stenosis surgery is associated with the collateral function [35–38]. Poor collateral circulation results in the maximal dilation of peripheral arteries of the MCA (reducing PR) through cerebral autoregulation to compensate for the flow into the MCA. In this situation, surgical dilation of the stenosis results in a significant increase in blood flow into the MCA, leading to CH. The results of our study are consistent with these conventional clinical perceptions and provide further quantitative evidence.

Based on the results of the UQ and SA, we determined that $\Delta \bar{Q}$ varied significantly with stenosis severity. The higher the severity of the stenosis, the more its dilation reduces the flow resistance of the ICA and the more drastic increase of the flow in this artery. Consequently, in patients with highly severe stenosis, the distribution of $\Delta \bar{Q}$ shifted to larger values (Fig 8).

Among the two measures of stenosis severity, $R_v$ was determined to affect $\Delta \bar{Q}$ more than $SR$ (Fig 11). $R_v$ is a measure of the viscous resistance of the stenosis, which reflects the axial diameter change and length of the stenosis (Eq (5)). In contrast, $SR$ is the stenosis ratio, evaluated using the diameters of the smallest and largest points. In clinical practice, $SR$ is commonly used to assess stenosis severity, as several criteria for diagnosis and treatment are defined based on $SR$ [1]. However, even if $SR$ remains the same, $R_v$ can vary considerably with respect to the stenosis geometry; consequently, the hemodynamic significance of stenosis can differ. Therefore, it is important to consider the viscous resistance that relies on the stenosis geometry along with $SR$ when assessing stenosis severity.

Remarkably, severe stenosis did not necessarily lead to CH ($\Delta \bar{Q} > 100\%$), as observed from the comparison of Patients 2 and 3. This is consistent with the clinical observation that there was no significant difference in $SR$ between the groups with and without CH [36,37]. When the stenosis is severe, the diameter of the arteries being the collateral pathway to the MCA on the stenosis side (i.e., ACA I, ACoA, PCA I, and PCoA) had a more significant impact on $\Delta \bar{Q}$ than the stenosis severity (Fig 11). A smaller diameter of the collateral artery limits the amount of collateral flow (Fig 10C and 10D), causing a compensatory decrease in the PR of the MCA (Fig 9C). Additionally, it prevents the increased flow in the ICA after surgery from being distributed to the six outlets, resulting in a large $\Delta \bar{Q}$ at the MCA where the flow is concentrated.

In Patient 2, the collateral flow decreased rapidly with the ACoA diameter <1 mm, and the risk of CH increased accordingly. This result supports the use of a diameter <1 mm to define the inadequacy of the collateral artery [75,76] and suggests that this criterion may apply to the risk of CH. However, the flow rate in one artery is affected by the geometry (particularly the diameter) of other arteries because the cerebral arteries form a ring-like network. Therefore, we believe that it is necessary to perform the UQ for each patient for reliable risk assessment, and the proposed approach is an effective tool for this purpose.

In summary, $\Delta \bar{Q}$ was intimately associated with the severity of the ICA stenosis and diameter of the ACA I, ACoA, PCA I, and PCoA that form the collateral pathway to supply blood to the MCA on the stenosis side. CH occurred when the following conditions were satisfied

 

simultaneously: (i) the stenosis was severe and (ii) the diameter of the collateral pathway was small.

## Limitations and future directions

The limitations of this study are summarized in this subsection. First, we assumed a uniform distribution for uncertain parameters without considering the possible differences in distribution owing to modality characteristics. The probability distribution of $\Delta\bar{Q}$ can change with the assumed distribution for uncertain parameters. However, in this study, we did not aim to predict the accurate distribution of $\Delta\bar{Q}$ but rather to conduct a rapid assessment of the possibility of CH ($\Delta\bar{Q} > 100\%$) by taking into consideration an intentionally wide range of uncertainties, so as not to miss any patient at risk. The results verified that the upper bound of $\Delta\bar{Q}$ is substantially lower than 100% in Patient 1 and slightly over 100% in Patient 3, which indicates that assuming different types of distribution (such as normal and log-normal) for uncertainties does not alter the probability of $\Delta\bar{Q} > 100\%$ significantly.

Second, the number of patients included in the prediction of CH was small. The results of the UQ and SA facilitated the clarification of the quantitative relationship between collateral circulation and CH. However, further validation is needed before the method can be used in clinical practice to assess the risk of CH. We believe that this can be achieved in the future with the availability of more patient data.

Finally, the proposed model ignores the peripheral collateral pathways [77] that cannot be acquired using existing imaging technology. The presence of collateral pathways other than CoW may contribute to preventing CH. In the future, we intend to focus on modeling the peripheral network in detail by integrating the patient geometry with mathematical models, similar to a previous study [78].

## Concluding remarks

Understanding the collateral function in cerebral circulation is essential for elucidating disease mechanisms and reviewing treatment options. In this study, the biology of collateral circulation in the CoW was explored by performing UQ and SA, which are the measures that stochastically evaluate the prediction result variability, using 1D–0D simulation that considers the entire cardiovascular system. The major challenge in performing these tasks in a clinical setting is its high computational cost. To address this problem, we constructed a machine learning-based surrogate model trained using the 1D–0D simulation data. The surrogate model accurately predicted the flow rate and pressure in the CoW while simultaneously reducing the prediction time to a few milliseconds. The results verified that the surrogate model enabled the execution of UQ with 100 000 predictions in a few minutes on a single CPU core and less than a minute on a GPU.

Leveraging the low computational cost of the surrogate model, we performed UQ in predicting the risk of CH, which is a life-threatening condition that can occur after carotid artery stenosis surgery if collateral circulation fails to function appropriately. Particularly, we predicted the statistics of the flow rate increase in the MCA after the ICA stenosis surgery, considering uncertainties in the parameters derived from the patient's clinical data. Furthermore, we conducted an SA to clarify the impact of each uncertain parameter on the flow rate increase. The results indicated that the flow rate increase was greater when (i) the stenosis was severe and (ii) the diameters of the ACA I, ACoA, PCA I, and PCoA that form collateral pathways to supply blood to the MCA were small. When these two conditions were satisfied simultaneously, the PR of the MCA on the stenosis side reduced significantly, and the flow rate increase exceeded 100%, i.e., that the surgery caused CH.

 

The proposed surrogate model can be applied more broadly to the prediction of the cerebral circulation and is not limited to the application demonstrated in this study. The approach facilitates the execution of computationally expensive tasks, such as UQ, SA, and extensive case studies. This can aid in analyzing the simulation results from a statistical perspective to gain new insights, accelerate the introduction of simulation tools into time-sensitive clinical practices, and facilitate translational medicine. Despite the existing technical limitations of large uncertainties in measuring cerebral circulation, the proposed approach explains the effects of uncertainties efficiently and helps in understanding various biological aspects of cerebral circulation, including its physics, physiology, pathology, and treatments.

## Supporting information

**S1 Appendix. Details on the design of experiments.**
(PDF)

**S2 Appendix. Details on the uncertainty propagation.**
(PDF)

**S1 Fig. Fully connected deep neural network used in this study.** (A) Network architecture.
(B) Schematic of a single node.
(TIF)

**S2 Fig. Arterial geometry of Patients 1–3.** Three-dimensional reconstructed arterial geometries obtained via lumen segmentation on computed tomography images.
(TIF)

**S3 Fig. $R^2$ scores of deep neural networks trained using different combinations of hyperparameters.** The number of training samples was maintained at 120 000, and the $R^2$ scores were evaluated considering 40 000 test samples. $N_{layer}$ denotes the number of hidden layers, $N_{node}$ indicates the number of nodes in each hidden layer, $N_{batch}$ represents the batch size for mini-batch training, and *lr* denotes the initial learning rate.
(TIF)

**S4 Fig. Scatter plots of flow rates (mL/min) obtained from the one-dimensional–zero-dimensional (1D–0D) simulation and surrogate model.** Flow rates in the carotid and cerebral arteries are depicted for 40 000 samples of test data. The negative flow rate indicates that the flow direction is opposite to that of the arrows in Fig 2. The $R^2$ score and mean absolute error (MAE) of each quantity are depicted in the corresponding panels.
(TIF)

**S5 Fig. Scatter plots of pressures (mmHg) obtained from the one-dimensional–zero-dimensional (1D–0D) simulation and surrogate model.** Pressures in the carotid and cerebral arteries are depicted for 40 000 samples of test data. The $R^2$ score and mean absolute error (MAE) of each quantity are depicted in the corresponding panels.
(TIF)

## Author Contributions

**Conceptualization:** Changyoung Yuhn, Marie Oshima.

**Data curation:** Changyoung Yuhn, Marie Oshima, Yan Chen, Motoharu Hayakawa, Shigeki Yamada.

**Formal analysis:** Changyoung Yuhn.

**Funding acquisition:** Changyoung Yuhn, Marie Oshima.

**Investigation:** Changyoung Yuhn.

**Methodology:** Changyoung Yuhn, Marie Oshima, Yan Chen.

**Project administration:** Changyoung Yuhn, Marie Oshima.

**Resources:** Changyoung Yuhn, Marie Oshima, Motoharu Hayakawa, Shigeki Yamada.

**Software:** Changyoung Yuhn, Marie Oshima, Yan Chen.

**Supervision:** Marie Oshima, Shigeki Yamada.

**Validation:** Changyoung Yuhn.

**Visualization:** Changyoung Yuhn.

**Writing – original draft:** Changyoung Yuhn.

**Writing – review & editing:** Changyoung Yuhn, Marie Oshima, Shigeki Yamada.

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
