## [Decision Letter · Decision Letter 0]

14 Apr 2022

Dear Dr. Yuhn,

Thank you very much for submitting your manuscript "Uncertainty quantification in the cerebral circulation simulation focusing on the collateral flow: Surrogate model approach with machine learning" for consideration at PLOS Computational Biology. As with all papers reviewed by the journal, your manuscript was reviewed by members of the editorial board and by several independent reviewers. The reviewers appreciated the attention to an important topic. Based on the reviews, we are likely to accept this manuscript for publication, providing that you modify the manuscript according to the review recommendations.

Sincerely,

Alison L. Marsden

Associate Editor

PLOS Computational Biology

Daniel Beard

Deputy Editor

PLOS Computational Biology

[LINK]

Reviewer's Responses to Questions

**Comments to the Authors:**

Reviewer #1: The authors propose a ML algorithm based on neural networks to efficiently estimate the patient-specific risk of cerebral hyperperfusion (CH) using uncertainty quantification (UQ). Clinical intervention is sometimes required in the presence of stenoses affecting the right or left carotid arteries. Cerebral hyperperfusion occurs when an increase of more than 100% in the circle of Willis flow rate is observed post intervention. The authors train a neural network to map 60 parameters describing the geometry of cerebral arteries and stenoses to 45 time-averaged quantities (flow rate and pressures). The gain in runtime with respect to 0D-1D reduced order models is dramatic (milliseconds vs minutes) and the possibility of parallelization makes this method attractive for UQ in clinical settings.

Overall, the methods discussed in this paper are scientifically sound and the strenght/limitations of the approach are clearly presented. Below are some minor remarks and observations that I believe should be addressed prior to publication.

1) Fig. 1 could be improved by including a brief description of the inputs (\\mathbf{x}) and the outputs (\\mathbf{y}_{sim}) either in the figure itself or in the caption; "sampling inputs that represent the anatomical and physiological conditions and collecting the corresponding simulation outputs" is too general.

2) Page 12, line 253: the authors mention that they use the Newton-Raphson method to enforce conservation of mass and total pressure at junctions. However, we use the Newton-Raphson to solve nonlinear equations, and these constraints are linear. I believe that the use of the Newton-Raphson method is necessary due to the presence of stenoses models which are nonlinear. I recommend clarifying this point.

3) Page 12, line 294: "within a reasonable range": the authors could add here that this aspect will be discussed later on in the paper (in paragraph Design of experiments). The reader might be confused by the use of "reasonable" here without further explanation

4) Page 13, line 301: there's an unmatched "(" in this sentence.

5) Page 13, lines 311-314: "The variation...as indicated in Equation (5)". These sentences are a bit unclear to me. Are the authors saying that Ls should also be varied because Rv depends on it, but they take it constant because the third term in Eq.(4) is negligible? If so, please rephrase to make this point clearer.

6) Page 17, Eq.(9) does the last layer feature a ReLu activation function? Isn't this incompatible with the type of normalization used for the outputs (standard normalization, discussed at page 19), meaning that negative values will never be predicted by the neural network?

7) Page 20, line 430: is Rv computed using Eq.(5)? If so, please refer to it for clarity.

8) Page 22, line 492: Please expand the title of the paragraph: SA -> Stability Analysis.

9) Page 23, Eq.(16): I am not sure that the upper bound in the sum is correct. If Nlayer = 1, the sum contains two terms as if the number of layers is actually two. Perhaps this is a problem of notation and Nlayer only refers to the hidden layers, whereas the authors consider the output layer a separate one.

10) Page 26: when discussing the parallelization, the authors could say a few words on how this was implemented. Are they launching one "simulation" at the time but using the GPU to optimize the matrix-matrix multiplications? Or are they launching multiple threads each performing a single simulation by exploiting the fact that the simulations are independent? In the latter case, it might be worth noting that the same could be done for the 1D-0D, provided that each process/thread have access to sufficient memory.

11) Fig. 8: what is the meaning of negative \\Delta \\overbar Q in each of the distributions? Should negative values be considered not physiological? Please clarify in text when discussing this plot.

12) Page 28, line 598: "The distribution of \\Delta \\overbar Q..." are the authors suggesting that more severe stenoses are associated with more uncertainty? Can they give an explanation of why this is the case?

13) Page 31, line 633: Can the authors explain more clearly what they mean by vertical and horizontal variations?

Reviewer #2: General comments:

This paper presents a novel systematic framework to evaluate collateral circulation in the circle of Willis (CoW) using a machine-learning-based surrogate model for blood circulation. The hemodynamic data in the surrogate model show reasonable correspondences with those in an original 0D-1D hemodynamic model, and the computational cost of the surrogate model is much less than that of the original hemodynamic model. This enables to reasonably perform uncertainty quantification (UQ) and sensitivity analysis (SA) focusing on some uncertainly geometrical and functional parameters in the analyses. Three patient-specific data are used for the UQ and SA, and risks of cerebral hyperperfusion are discussed with features of collateral flows in the CoW.

Since the approach is excellent and the obtained results and discussion are reasonable, I think the paper is worth being published in this journal. I nonetheless have a few unclear points listed below, so I would appreciate it if the authors clarify and discuss them.

Specific comments:

Page 14 – The authors state that the variation of stenosis length Ls is ignored, but also state that the effect of Ls is reflected in Rv in equation (5). This confuses me because the changing Rv is attributed to the change of Ls. Why did the authors directly vary Rv instead of Ls?

Page 14 – The surrogate model predicts cycle-averaged hemodynamic quantities, whereas the original 1D model provides a spatial profile in each vessel. Is the cycle-averaged in the surrogate model also indicate the spatial average in each vessel?

Table 2 – It is not clear to me what the boundary condition is imposed on the 0D-1D and surrogate model and how the value is varied in the UQ (what quantity does in Table 2 reflects the boundary condition?).

Page 19 – The authors normalize the training data being [-1,1] to improve the model performance. Is this a standard process for the DNN used in this study? I would appreciate it if the authors more clarify this point.

Fig. 4 – I could not understand the meaning of the “statistics converged” process in the “postoperative prediction” and the necessity for going back to the “Monte Carlo sampling” process under un-converged. Why does the post(-operative) process affect the pre(-operative) one?

**Have the authors made all data and (if applicable) computational code underlying the findings in their manuscript fully available?**

Reviewer #1: Yes

Reviewer #2: Yes

PLOS authors have the option to publish the peer review history of their article (what does this mean?). If published, this will include your full peer review and any attached files.

Reviewer #1: **Yes: **Luca Pegolotti

Reviewer #2: No

Figure Files:

Data Requirements:

Reproducibility:

References:

---

## [Decision Letter · Decision Letter 1]

7 Jun 2022

Dear Dr. Yuhn,

We are pleased to inform you that your manuscript 'Uncertainty quantification in cerebral circulation simulations focusing on the collateral flow: Surrogate model approach with machine learning' has been provisionally accepted for publication in PLOS Computational Biology.

Best regards,

Alison L. Marsden

Associate Editor

PLOS Computational Biology

Daniel Beard

Deputy Editor

PLOS Computational Biology

Reviewer's Responses to Questions

**Comments to the Authors:**

Reviewer #1: The authors addressed the comments in my previous review satisfactorily. Therefore, I recommend publication of the article in its present form.

Reviewer #2: Thank you for addressing the comments. I think the authors have addressed all of my concerns.

**Have the authors made all data and (if applicable) computational code underlying the findings in their manuscript fully available?**

Reviewer #1: Yes

Reviewer #2: Yes

PLOS authors have the option to publish the peer review history of their article (what does this mean?). If published, this will include your full peer review and any attached files.

Reviewer #1: **Yes: **Luca Pegolotti

Reviewer #2: No

---

## [Editor Report · Acceptance letter]

23 Jun 2022

PCOMPBIOL-D-22-00362R1 

Uncertainty quantification in cerebral circulation simulations focusing on the collateral flow: Surrogate model approach with machine learning

Dear Dr Yuhn,

I am pleased to inform you that your manuscript has been formally accepted for publication in PLOS Computational Biology. Your manuscript is now with our production department and you will be notified of the publication date in due course.

With kind regards,

Agnes Pap
